# A Systematic Literature Review and Meta-Analysis on Scalable Blockchain-Based Electronic Voting Systems

**DOI:** 10.3390/s22197585

**Published:** 2022-10-06

**Authors:** Uzma Jafar, Mohd Juzaiddin Ab Aziz, Zarina Shukur, Hafiz Adnan Hussain

**Affiliations:** 1Faculty of Information Science & Technology, Universiti Kebangsaan Malaysia, Bangi 43600, Malaysia; 2Center of Cyber Security, Universiti Kebangsaan Malaysia, Bangi 43600, Malaysia

**Keywords:** Bitcoin, Blockchain, cryptography, electronic voting, ethereum, Internet of things, hyperledger fabric, smart contract, scalability

## Abstract

Electronic voting systems must find solutions to various issues with authentication, data privacy and integrity, transparency, and verifiability. On the other hand, Blockchain technology offers an innovative solution to many of these problems. The scalability of Blockchain has arisen as a fundamental barrier to realizing the promise of this technology, especially in electronic voting. This study seeks to highlight the solutions regarding scalable Blockchain-based electronic voting systems and the issues linked with them while also attempting to foresee future developments. A systematic literature review (SLR) was used to complete the task, leading to the selection of 76 articles in the English language from 1 January 2017 to 31 March 2022 from the famous databases. This SLR was conducted to identify well-known proposals, their implementations, verification methods, various cryptographic solutions in previous research to evaluate cost and time. It also identifies performance parameters, the primary advantages and obstacles presented by different systems, and the most common approaches for Blockchain scalability. In addition, it outlines several possible research avenues for developing a scalable electronic voting system based on Blockchain technology. This research helps future research before proposing or developing any solutions to keep in mind all the voting requirements, merits, and demerits of the proposed solutions and provides further guidelines for scalable voting solutions.

## 1. Introduction

In many countries, technology plays a significant role in elections, which is necessary for some circumstances, such as the technology utilized to construct voter records, establish electoral boundaries, manage and educate employees, print ballot sheets, carry out voter education programs, cast ballots records, tally the votes, and compile and broadcast election results, to name a few. Technology can improve election organization, lower long-term expenses, and enhance political accountability if uses it correctly [1]. The technology used in elections can be as old as printing presses, ballpoint pens, or modern technologies such as computers and optical scanners, digital mapping, or the Internet; or it can be a mix of the two [1,2]. Organizing large-scale elections in countries without access to modern technologies can be difficult. It might change if Blockchain technology is the foundation for the electronic voting system.

For the sake of fostering public confidence and accountability among voters, electoral integrity is necessary not only in democracies but also in all societies [3,4]. From the government’s perspective, electronic voting technologies have the potential to boost voter participation and trust, as well as reinvigorate interest in the voting process. According to a new study, adopting electronic voting methods may improve security [5,6,7]. When deciding whether or not to utilize an electronic voting structure, it is necessary to research the factors that contribute to the system’s advantages over traditional voting methods, such as using paper ballots [8]. It is anticipated that it will improve the efficacy and efficiency of democracy; moreover, it will provide a solution to various highlighted problematic conditions. Likewise, it enhances election accessibility, lets the elderly and disabled vote, increases election turnout, and is simple to use while obtaining a quick result [9]. However, it is widely acknowledged that running electronic voting systems underneath tight security measures is essential, especially when using modern encryption methods [10]. In the beginning, electronic voting was proposed as a remedy to the issues caused by voting on paper ballots to ensure that elections would be fair and accessible [3]. Electronic voting system security problems have been widely researched in the literature. According to the findings of the research [11,12], electronic voting may present several challenges, including those related to data integrity, dependability, transparency, ballot secrecy, security, implications of fraud, repercussions of failure, ignorant voters, lack of special information technology skills, equipment storage, and cost.

Blockchain is a novel technology with much potential for information systems. Blockchain-based techniques are being researched in various fields, including healthcare, logistics, finance, etc. [13]. Electronic voting is becoming a more essential and widespread issue in the context of Blockchain and information systems. The unique characteristics of this technology, such as decentralization and immutability, were crucial in ensuring that the voting system followed the same norms as more conventional elections and voting fields [14]. This line of inquiry has gained traction recently due to rising allegations of electoral fraud, such as those levelled during the presidential election in the United States [8]. Democracy is founded on voting and will not work well if people do not trust the voting system.

On the other hand, it just takes a rumour to completely undermine people’s faith in a voting system, particularly an automated one. Conventional voting methods based on a paper presented a severe health risk during the COVID-19 outbreak; hence, the need for research on methods of electronic voting that are secure, impartial, and confidential became even more pressing [15,16]. Electronic voting systems’ reliability, speed, and security are not yet at a level that can boost the voter’s trust. Researchers are working on methods and procedures for a more secure and efficient electronic voting system to ensure anonymous voting is conducted in a fair and risk-free environment [17]. The implementation of Blockchain technology in voting systems has a reasonable chance of drawing the attention of researchers searching for the most effective solutions. Protocols can be implemented in a voting system to ensure that voting is conducted in a way that is private, public, and impartial [1]. Because of this, people’s trust in voting systems and democracy, in general, may increase.

Blockchain technology was the first and primarily used technology behind Bitcoin to keep track of financial transactions [18,19]. However, new proposals and applications have evolved in the past. Recently, the Blockchain-based electronic voting system has grown in importance to overcome some problems that might arise when voting electronically. Due to the immutable nature of the Blockchain, it has transformed into a decentralized and distributed voting system [20]. Consequently, Blockchain’s voting systems have been proposed as the next generation of electronic voting systems [21,22]. Because of Blockchain technology, governments are being driven to design voting systems that are both intelligent and bearable and incorporate information on sustainability into such voting systems. It ensures that all relevant parties access reliable information on long-term assets [23]. It is worth noting that even though Blockchain is increasingly being used to improve electronic voting system security, many difficulties remain in obtaining the full benefits from Blockchain. Identifying which problems should be addressed in designing a Blockchain-based voting system is required [24].

This research reviewed the literature and clarified such issues using a systematic mapping technique for this goal. This paper makes the following contributions: Identifying well-known proposals for scalable Blockchain-based electronic voting and their verification methods, and highlighting those proposals that used famous cryptographic solutions and their focus on cost and time. It also identifies performance parameters, the primary advantages and obstacles presented by different systems, and the most common approaches for Blockchain scalability. In addition, it outlines several possible research paths for developing a scalable electronic voting system based on Blockchain technology.

According to this study, Blockchain-based voting systems may solve data tampering and integrity problems. The most often expressed concerns with Blockchain voting systems are privacy, transaction speed, and scalability, which still need more work [25,26]. This article provides a systematic literature review of the motivations, research accomplishments, and applications intended to facilitate the integration of Blockchain and the electronic voting system. The main contributions are as follows:We describe the research methodology and a literature evaluation on scalable Blockchain-based electronic voting;We introduce the background knowledge of electronic voting and Blockchain technology from the perspective of scalability, summarized in the previous work. The motivations and benefits for applying Blockchain to electronic voting are discussed in terms of requirements and challenges in electronic voting and the characteristics of Blockchain;We summarize an optimized research framework according to the Blockchain architecture, including the scalability aspect of Blockchain and horizontal and vertical scalability trilemma in Blockchain, and analyse and specify significant characteristics of Blockchain scalability;As for the Blockchain applications in electronic voting, we reviewed the previous work on electronic voting based on Blockchain and focused on scalability;We analysed the previous research by following our methodology to answer the required questions regarding Blockchain function/performance requirements for electronic voting by comparing different Blockchain project implementations to provide a reference for practitioners;Finally, some open issues and challenges in the field of the electronic voting system, combined with Blockchain, are highlighted, such as shading, consensus algorithm, block size increase, directed acyclic graph, forking, and an increase in authorized hardware devices to decrease block generation rate;

This study was performed by PRISMA (preferred reporting items for systematic reviews and meta-analyses) [27]. It included Kitchenham’s [28] standard criteria to customize this SLR to the computer science area.

The rest of this paper is systematized, as depicted in Figure 1. Section 2 demonstrates an overview of electronic voting and Blockchain technologies. Section 3 debates the previous research closely related to scalability in Blockchain for electronic voting. Section 4 discusses SLR methodology. Section 5 focuses on our methods’ results and categories per the defined methodology. Research challenges and future directions are addressed in Section 6. In the end, we conclude this paper in Section 7.

## 2. Background Knowledge

This section is separated into the theoretical bases of Blockchain technology and electronic voting. A separate team is dedicated to discussing these characteristics in further detail.

### 2.1. Electronic Voting

Electronic voting is a method of voting in a political election or referendum that uses electronic technology in the voting process, as per the European Council [29]. It includes specialized electronic voting machines:Optical scanning;Electronically printed ballots;Centralized and decentralized software or applications for voting through the Internet;

To register voters, electronic voting machines make use of a variety of input devices. Some examples of these devices include keyboards and touch screens [30]. Paper trails, also known as voter-verified audit trails, are printed copies of recorded votes often presented for verification with the ballots themselves (VVAPTs). These devices are used for fast vote counting and collecting voter data. They also enhance ballot presentation, lowering the frequency of spoiled votes [29,31]. Some authors [32,33] believe that third parties, on the other hand, design specialized voting devices, making end-to-end verification rugged and lowering confidence in them. The optical scanners can scan and record votes on readable paper ballots. This method is simple to learn and provides quick and accurate results. In addition, this method utilizes paper ballots, which have not been tampered with but can be replaced and are very expensive to adopt and maintain [34].

Electronic ballot printers provide legible paper recipes or voting tokens that may be thrown away in vote boxes and counted by machines. Because of the tangible evidence, this strategy is transparent and provable [35,36]. However, this method is similarly costly, and the only benefit it has over the old voting system is that it prevents ballots from being spoiled. There are two main types of Internet voting software: centralized and decentralized. Both systems enable voters to cast their votes via electronic devices linked to the Internet [37]. This approach may take various forms, including specialized gadgets and web pages. Voting makes it easier for individuals to acquire rapid and precise results and makes voting more convenient [38]. Unfortunately, this mode of voting has the most severe security weaknesses, such as the potential for hacker attacks, inadequate anonymity and privacy, and the possibility of being forced to vote a particular way [39]. Two qualities may be used to classify electronic voting systems:Remoteness;Supervision.

Remoteness describes how the votes are transmitted for aggregation and counting. The votes are sent to a central counting authority in real time by a remote system using a communication medium such as the Internet [40,41]. On the other hand, a nonremote method gathers votes on the local level and then transmits them to a counting authority after the election has concluded [42]. The place of voting is determined by supervision. In a monitored system, votes may only be cast under the supervision of some authority, such as a polling station [43]. A nonsupervised method, on the other hand, permits people to vote from any place and cast their ballots. Electronic voting aims to enhance the conventional voting method by decreasing and controlling fraud, reducing human involvement, and speeding up result processing [44,45].

Moreover, expenses are lowered by concentrating on voting overhead and improving participation in democratic procedures by utilizing new technologies that are more accessible and useful [42]. Electronic voting methods, on the other hand, are not without flaws. The following are the most common issues that such systems face:Insufficient clarity and comprehension of such systems on the part of non-experts;An absence of standards and norms;Threats posed by system suppliers, malevolent users, and privileged insiders and the potential for these individuals to attack and manipulate the system;Cost increases due to required information and communication technologies (ICT) such as infrastructure, maintenance, and power consumption;

Various ways are used to address the issues in electronic voting, each using a distinct set of technology and algorithms. Blockchain technology has received much attention recently, and its potential for improving electronic voting systems has been acknowledged by [46]. The following parts will reveal the findings of this potential investigation.

In voting systems that use electronic equipment, ballots are either recorded or counted electronically. The term “electronic voting” refers to a voting process aided by computer gear and software. Such systems may handle various operations, from planning elections to storing vote ballots [47,48]. Laptops, tablets, and smartphones all fall under system types. The electronic voting system should contain voter registration, identification, voting, and tallying features.

The electronic voting system comprises the following processes: The first step is to create a voter registration list (registration). On election day, officials examine voters’ credentials (verification and authentication). People qualified to vote in the next phase may do so (casting collation). Encrypted and verified voting should be implemented in [49]. The votes must be kept secret, anonymous, and correct, and they cannot be changed or deleted [50,51]. Eventually, computerized voting systems can calculate votes simply by adding them by design (counting the presentation of results).

In general, computerized voting applications rely heavily on central authority control. Such systems have several flaws and hazards associated with them [52]. A few examples are the lack of electronic voting system norms, security and dependability risks, exposure to hacking, fraud susceptibility, malicious software development, machines’ high prices, and transactions’ safe storing [53,54].

The United States of America pioneered the usage of electronic voting in other countries in the year 2000, followed by France (2001), the United Kingdom (2002), Spain (2003), Ireland (2004), Estonia (2005), Portugal (2005), the Netherlands (2004, 2006, 2007), Paraguay (2008), Finland (2008), Austria (2009), Germany (2009), and Norway (2009) [3]. Estonia became the first country to employ electronic voting from a distant location in its national parliamentary elections of 2007 [55,56]. It came after the government used the technology in more limited legislative polls in 2005.

There are various voting methods available, each with its advantages and disadvantages. The most commonly used voting methods discussed in [57] are direct-recording electronic (DRE), punch card, public network DRE, kiosk voting, central count, and precinct count. It is essential to highlight that the characteristics listed in Table 1 should be included in all electronic voting systems.

### 2.2. Blockchain Technology

Blockchains are immutable distributed ledger systems that store data in a digital ledger without a central point of authority and failure. The defined consensus algorithm confirms every transaction on the public ledger on the system. Once a transaction has been made, it can never be changed [58]. Every node on the system is accountable for checking and verifying the system’s published data [59]. A platform made on Blockchain runs continuously on a peer-to-peer (P2P) network [60]. Anyone can connect to that network and receive incentives for running their software as a node. The first people who described and worked on the cryptographically connected chain of blocks were Stuart Haber and W. Scott Stornetta [61] in 1991. After that, in 1992, Bayer, Haber, and Stornetta [62] included Merkle trees in the design to increase the efficiency of collecting several documents into one block. The primary goal was to create a timestamp for digital copies to prevent manipulation. Satoshi Nakamoto [18] constructed the first Blockchain-based system in 2008. It is also important to point out that the cryptocurrency Bitcoin was the first prominent use of Blockchain technology [63]. The idea may be compared to a globally distributed open and secure data book. As a result, this technology may be used in the cryptocurrency and financial industries and other disciplines involving the transactions [64]. As a result, the idea is increasingly seen as a critical part of industry 5.0 applications in the future years. Although the Blockchain is famous in cryptocurrency, it is not unreasonable to believe that its possibility extends further to the digital currency [65]. Private companies and government institutions are also testing Blockchain [66,67].

“Blockchain” refers to a series of timestamped and cryptographically connected blocks. As other blocks are added, the total number of blocks in the chain will grow; the hash of the first block will be included in each subsequent block, as shown in Figure 2. Blockchain protects either secret or general data from being tampered with or manipulated [68]. The Blockchain is nothing more than a decentralized ledger that records all the direct transactions between users and vendors inside the system—a distributed network of nodes that maintains a shared transaction source [69]. The nodes are in charge of validating these transactions. As a result, the Blockchain enables trust to be established without a central authority [70].

Some have considered that Blockchain technology could ultimately be more significant than the Internet. Blockchain technology agrees that the data be saved and swapped on a peer-to-peer (P2P) network. Structurally, the Blockchain data can be conferred, shared, and linked with consensus-based algorithms [71]. It is decentralized, and there is no requirement for “trusted third parties” or mediators to be present along with the procedure [72]. The technology behind the Blockchain serves as the foundation for the new kind of Internet. Blockchain is derived from two theories: asymmetrical cryptography, which provides the use of a matched public and private key system, and distributed information technology architecture (especially P2P) [73].

Asymmetric cryptography is used in Blockchain systems, which is a slower kind of encryption than the symmetric cryptography [66]. A distributed system has components on various networked computers, which share and coordinate their activities by passing messages to each other. Asymmetrical cryptography allows users who do not know each other to transmit encrypted data [74]. This approach sends encrypted data to a third party using a public key that may be shared with anybody [75]. The third party decrypts the data using a private key associated with the public key. In other words, a public key is like a bank account number that anyone can share [76]. When users log onto their bank account, they must use the same password to access the private key [77,78]. Bitcoin was the first use of Blockchain technology. It is a digital currency based on Blockchain technology and can be used to deal online in the same manner as using money to trade in the natural world [79]. Because of the success of Bitcoin, Blockchain technology may be used in various industries and services, including financial markets, IoT, supply chains, voting, medical treatment, and storage.

Blockchain technology is broadly used in our daily lives, such as cybercrime, travel, healthcare, finance, and education. The technological community is also discovering other potential uses for this technology in other fields [80]. Applications such as insecurity, settlements, finance and assets management, banking, industries, and insurances are being used and evaluated by [81]. The Internet has modified how we share information and connect; the Blockchain will change how we exchange value and whom we trust. Many people in the Blockchain industry have noticed that Blockchain has become overhyped [82]. The technology has some limitations and is inapplicable for many digital interactions [83].

Current electronic voting technologies and solutions are unreliable, insecure, and hazardous. Blockchain technology in the electronic ballot has attracted many academics looking for considerably efficient methods and protocols for an electronic voting system that can perform safe, anonymous, and fair voting [84]. As a result, faith in electoral systems and democracy may grow. On the other hand, Blockchain-based solutions face several difficulties. As technology becomes more extensive and complex, it becomes more challenging to scale a Blockchain network [85]. This vulnerability affects small and large networks since there are only a small number of nodes, and a large amount of computing power is concentrated on specialized hardware [86]. Other attacks are also feasible, such as denial of service, Sybil, and man in the middle.

Last but not least, it is essential to note that anonymity is not automatically achieved by using systems based on Blockchain technology [87]. Because every node has access to a leading Blockchain network, transactions can be tracked to identify the primary owners of that transaction [88]. However, it is essential to mention that the noted obstacles (and benefits) are broad and may not involve every technological deployment.

There are other methods to classify Blockchain-based systems, but the most frequent one is based on the rights granted to users and nodes. These permits include:The ability to write;The ability to read.

Writing rights in a Blockchain network determine who is eligible to act as a node and take part in the storage of the Blockchain data structure, the negotiation of its scopes as parts of a consensus mechanism, and the distribution of mining rewards. Participants in the Blockchain must go through this procedure to agree on the field’s information [89]. Permissionless Blockchain networks allow anybody to act as a node and have full public access to these privileges. The phrase “permissioned Blockchain network” refers to a system in which only a few people can edit the ledger. On the other hand, read rights define who has access to a Blockchain’s data and under what circumstances [90]. These rights distinguish between public Blockchain networks, which enable anybody to read their contents, and private Blockchain networks, which allow only a limited number of organizations to view their contents [91]. Blockchain technology is evolving and improving all the time. There are several implementations available, each with its own set of attributes.

### 2.3. Scalability Aspect of Blockchain

Scalability means more transactions for the same hardware in this context, specifically the ability to increase the volume of transactions per second. Although Blockchain has seen substantial acceptance in recent years, one of the primary problems that may restrict its role as a disruptive technology is the scalability of Blockchain-based solutions [12]. As a result, this study aims to explore and assess existing initiatives to improve Blockchain scalability in electronic voting systems [92]. Our study has led us to conclude that scalability is a broad concept with various meanings in the literature. As a result, we use a definition of scalability from recent research to describe scalability in the context of this study, where scalability might relate to:Horizontal: This is accomplished by adding/increasing the number of computers in an existing pool/network;Vertical: This is accomplished by adding more power (such as memory, processing, and storage-efficient approaches) to an existing pool of resources. As a result, these core notions have been employed to describe scalability in Blockchains.

This study researched prior Blockchain scalability research, such as those mentioned in current surveys on Blockchain scalability and the fundamental scalability principles accessible in contemporary literature [93]. As a result, this section analyses and specifies significant characteristics that may be used to characterize a Blockchain system’s scalability. These characteristics of Blockchain scalability are described and presented in Figure 3.

### 2.4. Scalability Trilemma in Blockchain

Although Blockchain has become more attractive in recent years, one of the most significant issues with Blockchain-based electronic voting systems is scalability, which might restrict its potential as a disruptive technology. As a result, this study aims to look into and evaluate existing initiatives to improve the scalability of Blockchain-based electronic voting systems [94]. As a result, we use a definition of scalability from recent research to describe scalability in electronic voting systems in the context of this study. The scalability trilemma might relate to decentralization, security, and scalability [95]. It is essential to remember that the trilemma is only a model for the different obstacles that Blockchain technology faces [96]. The rule says that three elements cannot be combined at the same time simultaneously. So far, however, many researchers worldwide have experimented with many ways to optimize decentralization, scalability, and security.

Understanding the underlying scalability trilemma at the foundation of Blockchain architecture is critical. Blockchains suffer from a trilemma, as postulated by [97] Vitalik Buterin (cofounder of Ethereum) and further investigated by Multicoin [98] (an investment company that invests in Blockchain startups and crypto-related businesses as a “theory-driven investment firm”). They can only accomplish two of the following three features simultaneously. A Blockchain can have, at most, two of these three properties:*Scalability:* A high rate of transactions per second; decentralization: involvement of a vast number of individuals in the construction and certification of blocks;*Security:* Increasing the cost of gaining control over the network.

Scalability is not a priority for proof of work (PoW), a decentralized system that allows anybody to participate in mining or maintain a node. The cost of energy/hardware required to create accumulated hash power provides a security [99]. Just a few organizations control the world’s hashing power due to the proliferation of specialized and expensive Bitcoin mining hardware known as ASICs (application-specific integrated circuits) [100].

Decentralization and security are more critical to Ethereum than scaling concerning the second-largest network by market capitalization. Currently, the consensus process (PoW) is identical to that of Bitcoin. Its future proof of stake (PoS) mechanism will be built so that anybody may participate in block generation, just as in Bitcoin [101]. Proof of stake might be better than Bitcoin’s proof of work because the user does not need to buy costly equipment to participate in building blocks [102]. Consequently, the network will become more decentralized as the distribution of block producers becomes more diverse.

This study looked at current scalable Blockchain-based electronic voting systems, such as those included in the most recent surveys on Blockchain scalability or scalable Blockchain-based electronic voting, as well as the fundamental scalability concepts present in contemporary literature. As a result, this section analyses and specifies significant characteristics that may be used to characterize a Blockchain system’s scalability. This research defines these as other dimensions of Blockchain scalability trilemma in Figure 4.

#### 2.4.1. Decentralization over Scalability

Why have Bitcoin and Ethereum prioritize decentralization? Rather than prioritizing scaling, according to some observers, if these networks are to become widely adopted, they must be able to compete with Visa-like throughput. No one wants to utilize programs that take days, hours, or even seconds to complete a single task [103,104]. The criticism of Blockchains is legitimate, but the core benefits that Blockchains provide are missed, such as censorship resistance. Our return to traditional database systems offers censorship resistance. However, it comes at the expense of the efficiency provided by legacy systems such as Amazon Web Services. We will use traditional database systems if Blockchain platforms do not offer this functionality.

#### 2.4.2. Security over Scalability

The security of a Blockchain system is critical as a unique, promising technology aiming to build a name for itself by upgrading current infrastructure. Many crypto projects have prioritized decentralization and scalability above the security [105], with a slew of high-profile exchange breaches and manipulated source-code vulnerabilities. For all of its benefits, Blockchain ecosystems rely on the robustness of the underlying source code, which, like everything else, must be thoroughly scrutinized. Both decentralization and scalability will flourish if security is established; however, the decentralization process is ongoing, and scalability must be continually enhanced [106]. Blockchains have become an enticing target for hackers because the Blockchain source code is freely available to the public and has the potential for monetary benefit from a successful assault. While scalability concentrates on the positive aspects, security protects against the negative aspects, which are vital but often overlooked by [107]. Good Blockchain use cases have been inhibited by failures, such as the well-known decentralized autonomous organization (DAO) assault, which resulted from shoddy source-code security.

#### 2.4.3. Scalability over Security and Decentralization

Scalability leads to the ability of Blockchain technology to handle large transaction flow and future growth. Scalable Blockchains will not suffer as use cases multiply and Blockchain technology becomes more widely used [108]. The word “scalability” is used to identify Blockchains that cannot handle an increase in demand. According to the Blockchain trilemma, it is possible to achieve higher scalability, but this would come at the expense of either decentralization, security, or both fields [109]. Traditional, centralized systems offer faster network settlement times and higher usability than Blockchain networks [110]. Many Blockchain systems have achieved decentralization and security; however, scaling is a significant problem for today’s most popular decentralized networks, as shown in Figure 5.

Recently, there has been a growing interest in developing electronic voting systems via Blockchain technology. An overview of the most frequent answers found in the protocols, an introduction to current research in Blockchain technology, and an essential guide to the most prevalent challenges that the proposed solutions encounter are all possible outcomes of this article. The following is a list of the paper’s most important findings:
A comprehensive assessment of studies on scalable Blockchain-based electronic voting systems;Identifying the most critical trends in the subject;Identifying the key issues that scalable Blockchain-based electronic voting systems confront.

Between 2017 and 2022, this paper will research electronic voting systems based on scalable Blockchain technology. These databases IEEE Digital Library, Scopus, Springer Link, ACM, and Science Direct were searched to locate the papers used for the research.

## 3. Related Work

Blockchain technology has several uses in various industries, such as banking and finance, IoT and media, energy, health, logistics, and many more. In addition, new fields and ways of use are continually being investigated. The authors of [111] comprehensively evaluate several articles on Blockchain technology in information systems and give a comprehensive list of applications organized by Blockchain technology problems. This section provides a fundamental analysis of previous research that looked at scalable Blockchain technology for electronic voting. This research used digital libraries such as IEEE Xplore Digital Library, Scopus, ScienceDirect, SpringerLink, and ACM Digital Library to identify and review the existing literature to conduct a methodical study of such efforts. It allowed us to perform a rigorous study analysis of the efforts made. To identify existing surveys, this study focused on efforts to accomplish solutions for scalable Blockchain for electronic voting, as presented in Table 2. This investigation revealed many potential options [112,113,114]. These are broad studies of Blockchain technology, and they mention scalability as a significant challenge for Blockchain. These examinations do not specifically investigate or address the scalability of Blockchain technology, and as a result, they have been excluded from the scope of the present study.

Currently, research is being carried out on using Blockchain technology in online voting systems. The authors of [115] describe and compare numerous Blockchain-based electronic voting systems in their summary on voting. The deployment of Blockchain-based electronic voting systems [12] presents both threats and potential. Another significant survey was performed in [106], whose authors researched several Blockchain-based electronic voting systems to see how they adhere to existing international standards and conventions. To conclude, specific commercial Blockchain-based electronic voting systems are given and discussed in [116] in addition to an open electronic voting platform.

The authors of [117] investigated electronic voting obstacles and the present state of the Blockchain-based e-voting systems. According to [118], electronic voting may be unavoidable when comparing conventional voting with distant voting through the Internet. Finally, the authors of [119] present an overview of the current voting systems by describing several voting techniques, their benefits and drawbacks, and technical breakthroughs in the sector.

Numerous articles have been written on merging Blockchain technology with electronic voting systems. The results show how each method has distinct aims and is executed differently in these articles. These publications cover a variety of techniques to accomplish various objectives using a variety of ways. That is why we are performing an SLR on scalable Blockchain-based electronic voting to obtain a handle on what is new in the industry.

In conclusion, these studies lack both a systematic strategy for conducting the review and sufficient depth in current initiatives to address the scalability of Blockchain for electronic voting. Consequently, there are gaps in the coverage of several aspects of Blockchain scalability and the survey’s depth and breadth. Furthermore, due to the ever-increasing length of the Bitcoin chain and recent advancements such as Segregated Witness (SegWit), the scalability of Blockchain has recently garnered significant interest. Consequently, a current and extensive effort is necessary to systematically assess the scalability of Blockchain technology for electronic voting based on its present level of development. This evaluation aims to identify the work that has already been carried out, the existing limitations, and the potential avenues for future research. The first comprehensive literature assessment, based on [120], considers current initiatives that address all areas of Blockchain scalability in electronic voting systems by selected based on nine QAC discussed in Section 4.8. Some of the selected and related work is mentioned in Section 5 with the related studies, along with the references, research methods used, and a summary of their results, as well as pointing out the research gaps.

## 4. Research Methodology

We used an SLR to answer the study questions, and we followed the guidelines presented in [120]. Iteratively moving through the phases of planning, carrying out, and reporting on the review was one of the strategies we sought to use to conduct an exhaustive analysis of the SLR. The following sections explain the systematic literature review approach and an overview of the scalable Blockchain electronic voting studies.

### 4.1. Systematic Literature Overview and Process

A systematic literature review, also known as an SLR, is a secondary research method that follows a predetermined plan to locate, examine, and evaluate published research associated with a specific topic, problem, or phenomenon [120]. There are five stages to the procedure:Planning the review;Selection process;Conducting the review;Screening and refinement;Reporting the review.

Establishing research topics and defining a review strategy are the two goals of the review-planning phase. This step essentially presents the research’s scope. The primary goal of the review phase is to develop a search process, implement it, and analyse as many suitable primary papers as attainable. Following this phase, a collection of relevant articles will be picked from all of the available research papers. These papers may then be reviewed and used to answer the research questions. The last phase of the review is the reporting process, which entails composing the findings appropriately for presentation. This process ultimately produces the whole research report in a suitable format, which in this case, is a research paper. The steps of the systematic literature review are shown in Figure 6, together with the artefacts acquired after each phase.

### 4.2. Research Questions

By examining existing scalable Blockchain-based voting systems, this article intends to shed light on the most prevalent trends in electronic voting. To attain this goal, the research topics listed in Table 2 will be addressed.

As per our research, we interpret what we found in the literature and check if it is feasible, riveting, novel, ethical, and relevant to this research. During our examination of the literature, we ensured that the questions we produced were sufficient to demonstrate their relevance and that the techniques of analysis used were suitable.

### 4.3. Selection of Primary Study

To highlight the importance of primary research, specific keywords were categorized into a journal or search engine search feature. The keywords were selected to facilitate the discovery of study results that would contribute to answering the research questions. Moreover, “AND” and “OR” were the only operators used to limit this research. The search strings were as mentioned in Table 3.

Different search terms have been used to obtain the data in this research. This way, we did not miss any vital articles because some authors used variants of words to differentiate their work from others. This research tries to obtain and filter all the required data for data gathering, as shown in Figure 7. The platforms searched were:Scopus;ScienceDirect;IEEE Xplore Digital Library;SpringerLink;ACM Digital Library.

**Figure 7 sensors-22-07585-f007:**
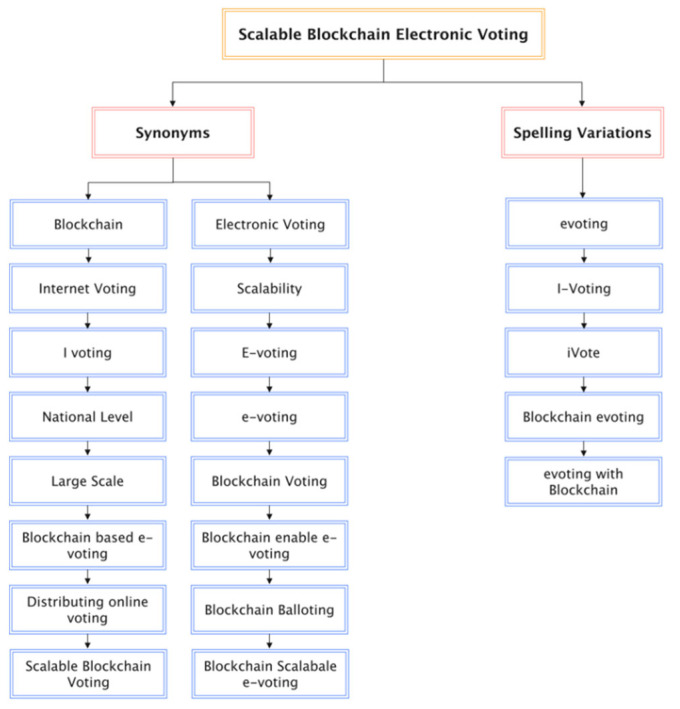
Terms that form the search string.

The search platforms were conducted using the title, keywords, or abstract, relying on the search platforms or databases. On the 31 March 2022, we conducted the searches and analysed all of the research that had been published up to that time. These searches yielded filtered results using the inclusion/exclusion criteria described in Section 4.4. The criteria enabled us to generate a collection of findings that we could then put via [121] snowballing process. Snowballing iterations continued until there were no more articles that fulfilled the inclusion criterion.

### 4.4. Inclusion and Exclusion Criteria

To filter the results of a database search, a set of inclusion criteria (IC) and exclusion criteria (EC) was developed in Table 4 and Table 5 include a comprehensive list. Studies that might be included in this SLR include case studies, uses of cutting-edge technology built on the Blockchain technology, and discussions on the progress of existing security processes via Blockchain integration. The included articles were peer-reviewed and published in the English language. Because Google Scholar may produce lower-quality reports, any results showing in Google Scholar were checked for compliance with these standards. This SLR has the most up-to-date findings from the research. The essential inclusion and exclusion criteria are listed in Table 2.

### 4.5. Strategy for Search

The following online libraries and archives were combed for information: IEEE Digital Library, Springer Link, Scopus, ACM Digital Library, and Science Direct. The study questions were prepared using the PICOC (population, intervention, comparison, outcomes, and context) [122] criteria in the review protocol.

*Population:* Articles describing scalable Blockchain-based electronic voting solutions for big or small-scale elections were reviewed.*Intervention:* Gathering information on electronic voting systems built on Blockchain platforms.*Comparison:* The results of the research will not be compared.*Outcomes:* It is essential to understand the scalability of Blockchain-based electronic voting systems and how they are utilized in real-world situations, advantages and disadvantages, and the cryptographic methods.Setting: Electronic voting, scalable Blockchain electronic voting, and Blockchain.

With this in mind, the following search phrase was eventually generated for searching the selected databases after numerous iterative tests: (blockchain OR block-chain OR distributed ledger) AND (voting) AND (large-scale OR national level) AND (scalable OR scalability) AND (lightweight).

### 4.6. Data Extraction

The search string to query the databases yielded 201 papers: 20 from Scopus, 34 from IEEE Digital Library, 83 from Springer Link, 22 from Science Direct, and 42 from ACM (Figure 8). This research was conducted by PRISMA (preferred reporting items for systematic reviews and meta-analyses). This process is shown in (Figure 9). For Scopus, IEEE Digital Library, Springer Link, ACM, and Science Direct, those percentages equate to 10 percent, 17 percent, 41 percent, 11 percent, and 21 percent, respectively (Figure 10). Figure 11 shows the graphical representation of publications obtained for each queried database per year. These papers went through a filtering process that was divided into four stages:The initial investigation, during which the majority of the relevant texts were gathered.Duplication removal, where removes the duplicate papers.The final selection was based on the title and abstract, and the inclusion and exclusion criteria were applied to the results.After a thorough reading, all chosen papers were subjected to inclusion and exclusion criteria.

In the first stage, 201 papers were produced, followed by 201 papers in the second stage, 101 papers in the third stage, and 76 papers in the fourth stage, as shown in Table 5.

**Figure 8 sensors-22-07585-f008:**
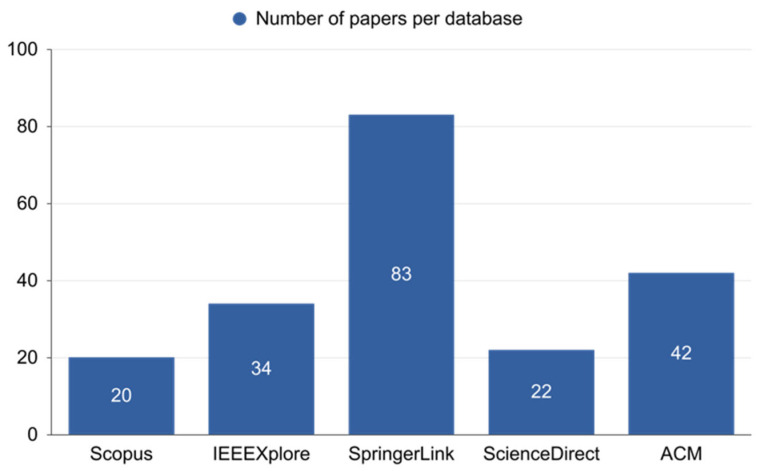
Number of papers obtained from each queried database.

**Figure 9 sensors-22-07585-f009:**
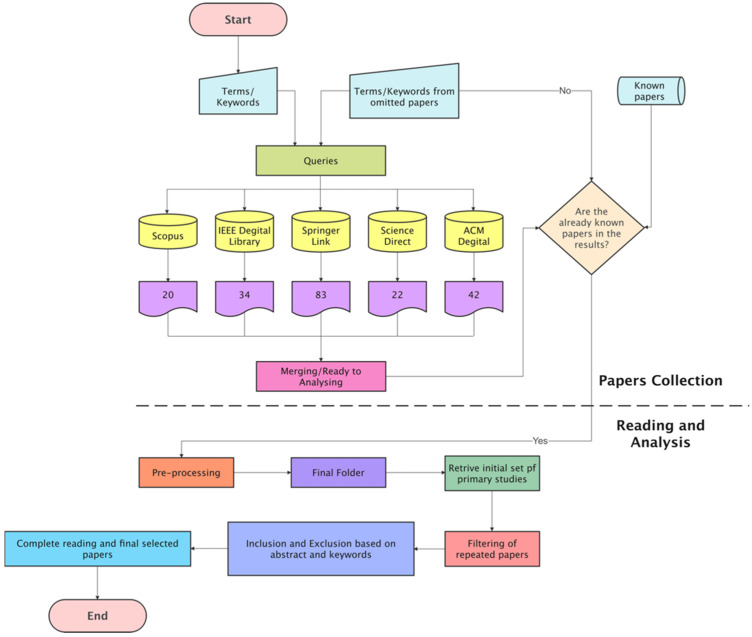
PRISMA systematic literature review process diagram.

**Figure 10 sensors-22-07585-f010:**
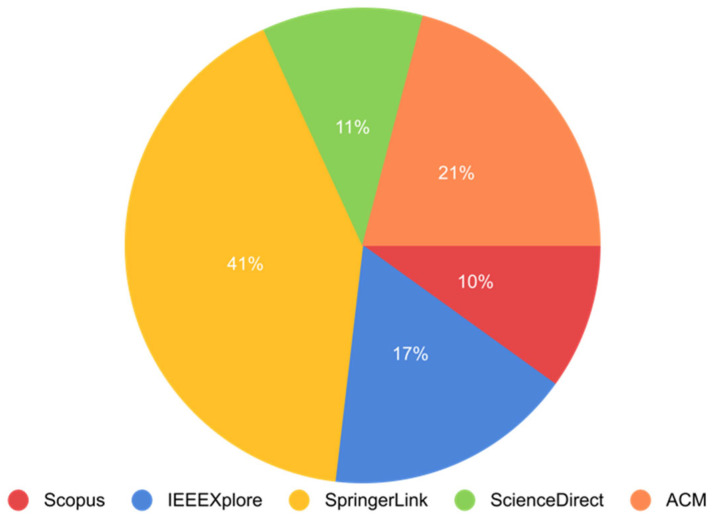
Percentage of articles found in each database searched.

**Figure 11 sensors-22-07585-f011:**
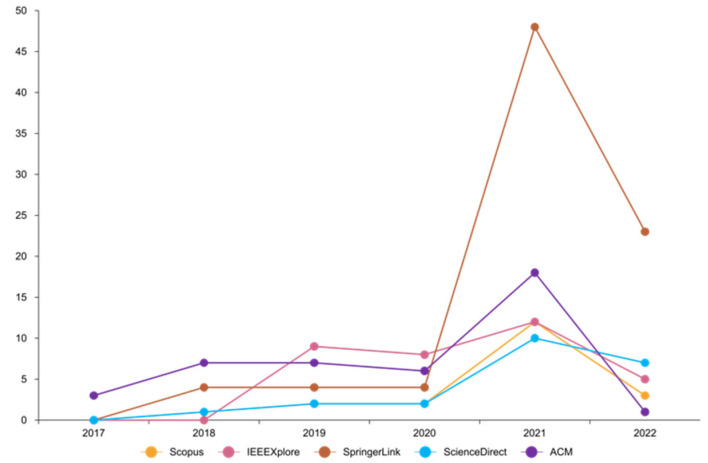
Graph of publications obtained for each queried database per year.

### 4.7. Data Analysis

Following the selection method, a framework for data extraction and synthesis of data was established. Following its implementation, data were extracted and synthesized from the 76 articles. Table 6 shows the form as well as its contents.

### 4.8. Quality Assessment Criteria

All studies shown in this paper were selected based on nine quality assessment criteria (QAC) presented in Table 7. QAC are used to refine and analyse the collected data. The presented paper satisfied the criteria to check the quality of the data collected for this research.

### 4.9. Compilation of Results

There are several challenges to the research’s validity. To begin with, not all relevant sources may have been discovered. The search was carried out across many databases using the most general search query feasible to counteract this. Thus, a substantial amount of research should have been conducted. The search was also undertaken to avoid discrimination and ensure internal validity. The use of well-known databases and the complete exclusion of grey literature ensured that the study’s external validity was maintained. As previously indicated, the complete search query was performed to collect the most relevant information from database search engines.

When undertaking a systematic mapping study, there are numerous threats to consider. An exhaustive search for relevant research and information may be impossible. This study identified several search parameters and investigated multiple databases to eradicate this threat. Using several criteria and logical operators could expand the coverage. This study sought to find all relevant documents using various keyword combinations. Even though the topic is new, most of the research conducted following the exclusion measures was published in 2017–2022. As a result, the missing paper review on the subject has too little an impact on these conclusions. Unpublished or associated works not obtainable in the chosen scientific database pose a threat. The internal validity is unaffected by the removed publications because the databases are well-known. Although this research encircled the articles following the designation criteria, there may still be a margin of error owing to the original sample.

## 5. Result Presentation

The findings of the systematic literature review of the selected 76 papers are reported in this section. Table 8 lists the papers that were chosen. Each research question is explored in its subsection. On the other hand, the primary subsection gives the findings of the research’s quality assessment.

### 5.1. Overview

Only studies from 2017 to 2022 were accepted for this study to observe the most recent research and ideas implemented in electronic voting based on scalable Blockchain technology. Table 8 shows the chosen article with benefits and years. All the selected articles were chosen based on the earlier criteria and then refined accordingly.

### 5.2. RQ1: What Are Some of the Most Well-Known Proposals/Implementations for Scalable Blockchain-Based Electronic Voting?

As previously said, there are a variety of Blockchain implementations. As a result, it is necessary to determine which one is currently investigated in electronic voting. Figure 12 shows which Blockchain implementations were found and how many times they were mentioned in the literature. Table 9 shows the relationship between specific articles and Blockchain framework implementations.

The first is [123], in which the authors claim that the hybrid consensus model ensures integral scalability and security. In addition to this, the second study [124] focuses on the system usability scale (SUS) test, which measures the efficacy, efficiency, and community satisfaction with the Blockchain-based electronic voting system. However, the systems are inappropriate in terms of authentications. In the third paper [125], which was initiated to ensure security using Blockchain and trust computation, the proposed mechanism shows a better success rate in simulation, but the accuracy did not confirm. In the fourth article [126], the authors look at how a transaction malleability attack may be executed. Even though the studies revealed the importance of variables such as network latency and block creation rate, the suggested approach has been implemented on a permissioned Blockchain instead [128], which advocates using permissioned Blockchain technology such as Hyperledger Fabric.

Furthermore, the authors argue that cryptocurrencies are inappropriate for electronic voting. Bitcoin is very famous in the financial sector and not in services, but it cannot deny the importance of Blockchain due to its first application. Here are some of the authors using Bitcoin in their research, such as [7,24,128,128,149,150,156,164].

**Table 9 sensors-22-07585-t009:** Studies utilizing specific Blockchain implementations.

#	Blockchain Framework	Articles
1	Bitcoin	[7,24,128,128,149,150,156,164]
2	Ethereum	[14,26,34,46,53,123,125,130,131,135,139,140,141,141,142,143,144,144,148,152,153,154,165,168,169,170,171,177,181]
3	Hyperledger Fabric	[29,46,129,136,145,146,158,160,163,185]
4	Hyperledger Sawtooth	[127,137],
5	Multichain	[126,149,176]
6	Helios	[165]
7	Phantom	[148]
8	Blockchain Database	[39,132,133,134,155,159,162,178,182]
9	Conflux	[147]
10	Waves	[157]
11	Platform Independent	[138,166,174,175,179]

In the services sector, with 29 articles, Ethereum is the first most popular Blockchain system, as below [123,181]. Ethereum is a decentralized platform that is open-source and can be accessed anywhere globally (Ethereum Foundation, 2014). It operates on both a public and a private Blockchain architecture. Enabling smart contracts, also known as programmable contracts, can automatically carry out their terms between two parties. Because it is used in all studies, this feature is critical for choosing this Blockchain implementation. It is worth emphasizing that Ethereum is a proof-of-work system, with each transaction on the Blockchain costing gas. It suggests that any system based on this technology ought to take into account any expenses incurred as a result of using it. These systems may not be suitable for large-scale elections because of the possible costs or the added strain that comes with managing virtual currency. The Linux Foundation is the organization in charge of managing the Hyperledger commercial Blockchain project.

It is a worldwide corporation that provides the essential structure, rules, norms, and tools for developing open-source Blockchain and associated applications for usage in various sectors. It is meant to facilitate pluggable implementations of diverse components and handle the complexities and subtleties that exist throughout the economic ecosystem. The above authors [163,185] used Hyperledger Fabric for private Blockchain, although Fabric is much faster than the rest of the above Blockchain frameworks, but the network is not decentralized in any way. In the vast majority of cases, it is controlled either by a single entity or by a group that has shared responsibility. Two different motivations led to the decision to concentrate some portions of the database. First, it is managed by a group, and second, it is often hosted on a centralized cloud service provider such as Amazon Web Services or Microsoft Azure.

### 5.3. RQ2: Were Those Solutions Tested in a Real-World Scenario?

This question tries to answer if any of the 76 possible solutions explained in the 76 articles chosen were used in an actual election or referendum, whether the results impacted the final product, and how that impact would have been felt. Some of the scenarios were just tested and simulated to check the feasibility of the mentioned solutions. The above solutions have been implemented in some states, but not on a large scale or country level. Some small organizations implemented internal voting, and they found some flaws presented in these solutions and needed more research to obtain fully controlled and decentralized results.

### 5.4. RQ3: What Are the Verification Methods Used to Test Those Solutions?

This question aimed to find out what testing procedures were used to verify e-voting systems. Table 10 shows a list of the test techniques that have been identified, along with their publications. The analysis of assessment criteria is a test method designed for writers to officially and informally evaluate how well their proposed systems meet a set of requirements. It is a form of verification that is used in the majority of publications.

Scalability testing is a type of testing through which the developers perform a formal and informal investigation of the provided solution’s capacity to manage many users and transactions simultaneously, such as those present during an election. One example of this testing scenario is when voters cast their ballots online.

Performance tests are a type of method test in which the authors verify the performance of their solution in the real world. It involves load testing and monitoring the time it takes to complete various election processes, such as voter registration, setup, and ballot casting and tallying.

Security analysis is a testing method through which authors formally and informally analyse and define the security qualities of their systems in response to specific attacks such as reply attacks, DDoS attacks, Sybil attacks, and man-in-the-middle attacks. Examples of these attacks include reply attacks, DDoS attacks, and Sybil attacks.

Cost evaluation is a testing approach in which writers assess the expenses associated with using their product. It is essential when the Blockchain platform is open to the public and employs a cryptocurrency such as Ethereum, where each action has a cost measured in gas [151,159] conducted three types of tests, while only [131,183] ran four types.

### 5.5. RQ4: What Are the Different Cryptographic Solutions Employed in Previous Research?

The goal of this topic was to find the more often utilized cryptographic solutions in present electronic voting research. Table 11 shows the results linking cryptographic solutions to specific research articles.

The Advanced Encryption Standard (AES), also often referred to as Rijndael, is a standard for the encryption of electronic data established in 2001 by the National Institute of Standards and Technology in the United States. Around the globe, critical data stored in software and hardware are encrypted using AES. It is essential for the government’s data security, cybersecurity, and computer security [127,134].

Homomorphic encryption is a kind of cryptographic solution that enables the generation of ciphertext to perform computations that provide the same result as if those computations had been performed on the plaintext. This type of encryption is known as symmetric-key encryption. Homomorphic encryption is employed to encrypt votes as mentioned in these articles [128,179].

The Rivest–Shamir–Adleman (RSA) algorithm is an asymmetric encryption technique that encrypts and decrypts data using two independent keys known as public and private keys. When using RSA, sensitive information may be encrypted using a public key, and the encrypted message can only be decrypted using a private key [137,160].

Elliptic curve cryptography is a key-based encryption technique, sometimes known as ECC. The encryption and decryption of data sent over the Internet is the primary emphasis of the ECC protocol, which makes use of distinct public and private key pairs. Concerning ECC, the cryptographic technique known as Rivest–Shamir–Adleman (RSA) is often discussed [24,162].

The elliptic curve digital signature algorithm, often known as ECDSA, is a kind of digital signature algorithm (DSA) that generates signatures using elliptic curve cryptography keys (ECC). This equation is based on public-key cryptography, and it is quite effective (PKC) [129,131].

As can be seen, the cryptographic solutions that are utilized the most frequently include a variety of digital signatures (12 articles) for authentication and authorization, ElGamal (4 articles) for anonymity and privacy, and DSA encryption of various types (1 article) for operations on encrypted votes, such as counting. The SHA (secure hash algorithm) (14 articles) is an algorithm that is extensively used in security protocols and applications. Some examples of these include transport layer security (TLS), pretty good privacy (PGP), secure sockets layer (SSL), Internet protocol security (IPsec), and secure/multipurpose Internet mail extensions (S/MiME).

### 5.6. RQ5: Were the Cryptographic Operations Used in Prior Solutions too Costly and Time-Consuming?

The goal of this question was to find the cost and time of the current cryptographic solutions in the latest research regarding scalable electronic voting systems. Table 12 shows the results, linking cryptographic cost and time to specific research articles. The United States government protects sensitive information via encryption known as the Advanced Encryption Standard (AES), a symmetric block cipher. Encryption of sensitive data all over the globe is accomplished with the help of the AES algorithm, which is implemented in both software and hardware. It is essential for the preservation of data and the security of government computers and cybersecurity, [127,134]. AES focuses mainly on the four steps used in each round of AES encryption as follows: (1) byte substitution, (2) shift rows, (3) mix columns, and (4) add round key.

AES is at least six times quicker than triple-DES (Data Encryption Standard). A substitute for DES was required as its key size was too small. AES is a symmetric algorithm designed for private-key cryptography. It is faster than RSA but only works when both parties transmit a private key.

Homomorphic encryption techniques are a kind of encryption algorithm that permits users to conduct mathematical operations on encrypted data. It is a beneficial property with several applications [39,128,133,151,159,165,179]. Currently implemented completely homomorphic systems are several charges of magnitude slower than unencrypted data operations. Homomorphic malleability is one of the various theoretical issues. It indicates that homomorphically encrypted data can be converted to another type of encrypted data.

The Rivest–Shamir–Adleman or RSA algorithm is the cornerstone of a cryptosystem. A cryptosystem is a set of cryptographic algorithms for certain security services or goals. It enables public-key encryption, which is used extensively to protect sensitive data, particularly when such data are sent over an unsecured network [137,160]. Due to the massive number of calculations, RSA is somewhat slow and costly. It is still slow in contrast to symmetric encryption techniques.

Elliptic curve cryptography, sometimes known as ECC, is a method that encrypts data using a key. Pairs of public and private keys are emphasized heavily in ECC’s approach to the decryption and encryption of online traffic. The cryptographic technique known as Rivest–Shamir–Adleman (RSA) is often cited in connection with ECC. Elliptic curves provide the same level of protection against intrusion as conventional security methods (such as RSA), but they do so with fewer bits. Elliptic curve cryptography requires, among other things, a decrease in the size of the device, a reduction in the amount of power that is used, and an improvement in processing speed [24,162].

ECDSA is a kind of DSA that makes use of keys obtained from ECC. This equation, which is based on public-key cryptography, is beneficial due to its efficiency (PKC). ECC is used to produce keys, resulting in far more minor keys than the typical key generated by digital signature [129,181]. Some studies claim that ECDSA is more efficient than RSA when signing and decrypting, but that it is slower when verifying signatures and encrypting data. In comparison to other cryptographic systems, the ECDSA provides significant benefits. With smaller key sizes, it gives more security.

The Diffie–Hellman key exchange is the foundation for the ElGamal encryption system, an asymmetric key encryption strategy that uses public-key cryptography. Taher Elgamal first described it in 1985 [132,162]. It is harder to factor huge prime numbers in RSA than in ElGamal, which relies on discrete logarithm calculations. RSA has been demonstrated to be more efficient than ElGamal in encrypting data. In contrast, the ElGamal decryption procedure is far quicker than RSA’s.

The DSA has been designated the Federal Information Processing Standard for digital signatures. Using the modular exponentiation and the discrete logarithm issue both come from the subject area of mathematics. DSA is faster at decryption and signature, but RSA is better at encryption and verification. When dealing with performance concerns, it is good to consider where the problem is (i.e., whether it is the client- or server-based) and then decide based on the basic methodology [144].

SHA is an acronym for a secure hashing algorithm that is safe and reliable. Data and certificates are hashed using SHA, a newer version of MD5. This algorithm, known as a message digest, results in a 160-bit (20-byte) hash value known as a message digest. Sensitive data may be protected via hashing and encryption. On the other hand, passwords should seldom be encrypted but rather hashed. The hash algorithm can only be used in one specific way, and that is: (it is not feasible to “decrypt” a hash in order to recover the value of the plaintext that it was initially being encrypted from). It is a good idea to use hashing to verify passwords, whereas encryption is used to secure users’ data, and it can be decrypted [182,185].

### 5.7. RQ6: What Are the Latest Blockchain Applications Focused on Scalability?

The selected articles can be categorized into three groups, as shown in Table 13: the first is the small-scale or organization voting group, which consists of 20 articles including [183,185] that propose solutions and methods for small-scale voting, such as boardroom voting, etc. National voting is the second group, which consists of 24 papers [123,167] that describe e-voting on a large scale such as national elections, but these solutions are not tested in real-world scenarios; some of the solutions are verified formally and some of the solutions are tested by performing simulations. The third group comprises 18 publications [135,161] that provide solutions for generic voting. This category contains papers that did not specify the sort of election.

### 5.8. RQ7: What Parameters Test the Performance and Scalability of the Electoral Process on a Large Scale?

Table 14 shows the parameters that have been identified to check the performance, highlighted in bold. Increases in block frequency may improve performance [140,178], while block propagation time sets a lower limit. The related work contradicts itself considering the effect of block size on performance, while [24,185] claims a favorable effect of increasing block size on performance. Depending on their instructions and storage access, different smart contracts may have various runtimes in terms of workload [2,48,134,136,141,142,143,144,146,147,151,156,157,159,162], Furthermore, [153,165,168,171,177], for example, reports a performance disparity. Node configuration and CPU usage are the parameters that describe a node’s computing power; they comprise hardware settings such as the quantity and kind of CPU and RAM. Only a tiny amount of research was conducted on this parameter in [14,34,179]. The impacts of network size are discussed in only a few studies [181,186].

Network structure/network usage is a parameter that describes the structure of the Blockchain network. This research is found in some of the work conducted in these papers [125,182]. Workload quantity is the number of transactions processed within a given time. Regarding workload quantity, the related work provides contradictory statements, such as in [1,165], while the quantity of miners/sealers/memory usage refers to actively participating nodes to be handled within the given time. Some of the work conducted on these papers was discovered during this investigation [14,164]. Blockchain client and API are referred to, as it is a user interface that connects to a Blockchain node or client network directly or via another service. During the initial investigation, some of the work on these documents was discovered clearly in these papers [136,166].

Subsequently, after applying the refinements to determine scalability parameters, 51 articles were picked to be retained in our study spanning three subcategories as follows. The first subcategory, throughput, includes 15 publications (Table 15) that look at scalability issues. A fundamental component of scalability is transaction throughput, and almost all studies use the term “transactions per second” to describe throughput (TPS). It is the speed at which valid transactions are committed and added to the block when the stakeholders in a Blockchain network (miners) agree on how the network should function. It is worth noting that only these authors discuss the throughput [123,178].

Meanwhile, the rest focuses on the second subcategory of scalability latency. In computer science, the term “latency” refers to the time that elapses between an input and an output. In Blockchain technology, having a low network latency is essential. Latency can discuss two types of delays in Blockchain: network latency and transaction latency. The network latency occurs between when a transaction request is initiated and when the network confirms the transaction. A transaction’s latency is a statistic of consensus efficiency that affects the processing and execution of large numbers of transactions. The second subcategory consists of 17 publications providing solutions for scaling the Blockchain by improving latency, such as [126,163]. In the following 19 articles [147,177], when modifying a parameter, these articles utilize throughput and latency instead of changes in throughput and latency (e.g., the network size or the hardware configuration of a node).

### 5.9. RQ8: What Are the Prior Approaches for Blockchain Scalability to Efficiently Enhance the Electoral Process on a Large Scale?

The goal of this question was to highlight approaches for Blockchain scalability to efficiently enhance the electoral process on a large scale. Table 16 presents a list of identified methods with their publications. On-chain methods boost the scalability of Blockchain by altering internal settings to reduce network latency and optimize the number of transactions and messages, respectively.

On-chain solutions include: Blockchain pipelining—an on-chain approach that adds blocks to the main chain without other nodes’ validation. It improves the transaction throughput of the Blockchain. The final decision on the block’s validity for forming the main chain is made through voting among nodes as a separate layer [145,187]. Blockchain delivery network—this research investigates solutions [126,164] that use cut-through routing-enabled gateways or cloud delivery networks. These ideas try to improve the transaction throughput or storage scalability without disrupting the Blockchain’s decentralized character. Block size adjustment—adjusting block size is another scalable method. These procedures are application-specific and require adjustment. Too many increases in block size can enhance the transactions per block, but latency is propagated. On the other hand, too much of a reduction in block size can raise block generation rate (BGR), improve latency, and cause frequent forks [1,188].

**Off-chain solutions:** Off-chain solutions carry out transactions outside the primary Blockchain network, reducing the required effort. These solutions are powered by Blockchain technology. Off-chain technologies such as payment channels link (LN), Raiden Network Token (RDN), or sharding may increase a Blockchain network’s horizontal and vertical scalability. These solutions have the potential to provide Blockchain-based solutions for devices with limited resources, such as Internet-of-things devices.

*Payment channel networks:* By constructing a micropayment channel, the payment channel allows several parties to conduct various off-chain transactions without publicly committing all transactions. Minimizing the workload on the main chain leads to an increase in throughput. In a typical payment channel network, just two transactions are required to update a record on the main-chain to complete all transactions between parties or satisfy the need for an on-chain transaction. Participants in this network can undertake an unlimited number of transactions. Through intermediaries, even parties not in a direct relationship can enter into transactions [128,160].*Sharding:* A Blockchain’s mining node stores all the states, including account balance and transaction history, which reduces transaction throughput linearly. Sharding divides an extensive database into manageable pieces to boost efficiency [123,127]. In the Blockchain, it is the horizontal separation of the main chain into shards. Each partition/shard stores its state. Sharding is a technique that separates the main chain into multiple independent groups, even though it is considered an off-chain solution. As a result, each transaction broadcast on the network does not have to be mined by a single node. Each shard acts as its Blockchain inside the network through the Merkle tree and may be joined to the main chain using cryptographic means.

**Hardware-assisted approaches:** In addition to software-based approaches, the body of published research includes several solutions that employ specialized, trusted hardware devices for one of two purposes: either improving consensus or speeding up the transaction process to improve Blockchain scalability [129,146]. The hardware has high processing machines and trusted execution environments (TEEs), which enable efficient transaction management while maintaining both correctness and speed.

**Parallel mining or processing:** The conventional implementation of Blockchain technology is based on decentralized mining. Additionally, transaction throughput and scalability are severely limited in this implementation. The parallel mining methods improve the scalability of Blockchain networks by mining several blocks concurrently without making any fundamental changes to the Blockchain’s underlying structure [147,187].

**Redesigning Blockchain:** Through this research, the study discovered a few ways to devise an effective strategy, such as to design a new consensus structure, to deal with numerous properties of scalability such as increasing throughput and reducing latency etc. [152,185]. Although Graphchain and HashGraph leverage the nonlinear creation of blocks through the usage of directed acyclic graphs (DAG), certain techniques present as an alternate (other than Blockchain) scalable distributed ledger (DLT) solution. However, these methods are out of the scope of the current research.

**Table 16 sensors-22-07585-t016:** Prior approaches to enhance electoral process.

Scalable Classifications or Approaches		Impact on Scalability	Articles	Issues/Challenges
**On-chain**	Blockchain pipelining	Throughput	[145,187]	It is well-known that some approaches, such as BigchainDB, are vulnerable to a 33 percent attack.
	Blockchain delivery network	Throughput and storage	[126,164]	Malicious actors may be able to spread blocks via this vulnerability. Reliable network infrastructure is a must here.
	Block size adjustment	Throughput and latency	[1,188]	Application-specific, forks, increased block generation rate (BGR)
**Off-chain**	Payment channel networks	Throughput	[128,160]	Issues with one’s privacy and security. For transactions to be committed, it is necessary for both parties to be online or to lock their tokens.
	Sharding	Throughput	[123,127]	Security (shard takeover problem) Exchange of information (maintaining atomicity and preventing overloading of shards in cross-shard transactions)
**Hardware-assisted approaches** **Parallel mining** **Redesigning a new Blockchain**		Throughput and latency	[129,146]	Adding nodes degrades performance. TPS/latency centralises. Requires an attractive incentive system
		Throughput and BGR	[147,187]	Requires numerous active miners in a single network
		Horizontal/vertical scalability	[152,185]	Fork avoidance, adversaries, avoidance incentive mechanism

## 6. Analysis and Discussion

According to researchers, electronic voting systems are expected to benefit from the use of Blockchain technology. The immutability of data and the system’s distributed nature are the primary benefits. Researchers have paid close attention to the issue of Blockchain’s scalability since it has emerged as a significant concern. We need ways and processes to increase horizontal and vertical scalability due to the growing usage of Blockchain technology in financial and nonfinancial industries. When scaling a Blockchain for large or countrywide elections, the most important pinpoints to consider are simplified storage, increased throughput, and reduced latency. Decentralized, secure, and scalable (DSS) Blockchain applications or solutions have been presented in various ways, each with its advantages and disadvantages. Blockchain cannot achieve its full potential unless the scalability issues are addressed. As a result, we have identified gaps in the current state of the art that will necessitate more work from the scientific community. In order to build scalable Blockchain, we have outlined the fundamental research problems below.

### 6.1. Sharding

Sharding is a method that may be used to increase the scalability of Blockchain. Graphchain [189] and Omniledger [190] are two examples of sharding-based techniques suggested to achieve low-level latency and high-throughput in a distributed database. Both of these techniques were developed in 2018. There are many other ways to solve such problems, but Graphchain has emerged as the most efficient and secure one [191]. However, these methods only apply to cryptocurrencies that do not require permissions. On the other hand, some other approaches were extended to all workloads but depended only on trusted hardware to reduce communication overhead [192].

In particular, message complexity reduction is the aspect of sharding and Blockchain sharding, which needs further investigation. Blockchain’s scalability depends on low communication costs per transaction (CCPT) [193]. CCPT is regarded as scalable in the context of Blockchain, either relying on hardware specifically designed for the purpose or trusting that all nodes are interested and would act rationally to sacrifice reliability to gain CCPT. Implementing decentralized reputation-management systems and load-balancing mechanisms that include an attractive incentive strategy are potential areas of investigation that could be pursued to prevent anomalous behaviour from nodes that do not have dedicated hardware [194]. Both of these areas could be explored further. Both of these areas are intended to prevent anomalous behaviour from occurring.

### 6.2. Consensus Algorithm

Due to the decentralized nature of Blockchain, the consensus algorithm is an essential part of every software stack. Even though Bitcoin is built on proof of work, various additional consensus algorithms have been suggested, comprising proof of stake, proof of authority, proof of weight, etc. An empirical investigation of consensus algorithms is necessary to understand and emphasize the applicability for various application domains [195]. This consensus algorithm may be a new topic of study that needs additional investigation by the scientific community.

### 6.3. Block Size Increase

As a result of the study, this research can identify the function block size and generation have in scaling Blockchain. Increasing the maximum block size is known as the block size increase. The blocks in Blockchain networks are created regularly, including a record of all transactions. Because the number of transactions that may be stored in a block is limited by the block size, raising the block size will boost throughput [196]. It can cause unacceptable propagation delays for blocks if the transmission delays caused by larger blocks are too significant. However, most of these attempts have focused on the Bitcoin Blockchain and are consequently limited to the Bitcoin settings. Such an examination should be carried out on a more abstract platform such as Ethereum or Multichain to emphasize the benefits and weaknesses, especially scalability. It is hoped that this will help developers understand the importance of Blockchain parameters and enable them to select appropriate Blockchain platforms for various application areas.

### 6.4. Directed Acyclic Graph

Another Blockchain form distinct from regular Blockchain is the decentralized autonomous group or DAG. It is a network consisting of separate transactions connected to several other individual transactions [197]. The DAG is a tree that grows from one transaction to another, branching out from one transaction to another, etc. Blockchain is a connected list of blocks.

### 6.5. Increase Authorized Hardware Devices to Decrease Block Generation Rate

Using innovative or approved hardware equipment for mining and verification on a permissioned (consortium) Blockchain may slow the pace at which blocks are generated, directly influencing the amount of business conducted on the network [198]. On the other hand, the use of this technology in permissionless (public) Blockchain systems requires an effective incentive mechanism to persuade miners to employ full gear with increased processing power, storage space, and memory.

## 7. Conclusions

The main aim of this paper is to review and analyse the current research on scalable voting systems, primarily based on Blockchain technology. Nevertheless, developing and implementing an electronic voting system is not a simple undertaking. Electronic voting systems must solve many problems, such as authentication, privacy, data integrity, transparency, and verifiability. This report is a systematic mapping study that summarizes the current research on scaling Blockchain technology for electronic voting. One of the fundamental issues that prevent the general use of Blockchain technology in various application fields, including electronic voting, is the impossibility of scaling Blockchain to accommodate increasing numbers of users. Recent occurrences, such as the epidemic that has spread all over the globe and the rise in the number of instances of election fraud, have resulted in the need for a voting information system that is effective, scalable, safe, and dependable. Because of this, investigations into new and improved Blockchain-based solutions are still being carried out. The primary objective of this research is to shed light on the various Blockchain-based voting options currently available. The paper presented the first systematic effort to identify and collate existing efforts related to Blockchain scalability in these aspects. It includes leveraging Blockchain to achieve scalable applications, mechanisms, and methods to enhance Blockchain scalability by contributing to the core Blockchain functions and efforts to define the scalability challenges through analysis of Blockchain-based electronic voting solutions. Several study holes in the field of elevating have been provided that need to be considered for further investigations. There may be other downsides, such as resistance to compulsion, scalability assaults, reduced transparency, and untrustworthy systems, all of which need to be overcome. Testing the electronic voting system may be conducted in various ways, much like trying any software, including acceptability, performance, and security. On the other hand, there is no universally accepted method for checking and ensuring such systems’ accuracy or reference data. At the very least, no one is referenced in any considered books. In addition, there is no indication that these technologies are being utilized in reality, making it hard to conduct an exhaustive study. Most of the chosen papers provide verifiable answers, one of the primary challenges of the electronic voting method. Other concerns often addressed include protecting the confidentiality of ballots and determining who is eligible to vote. On the other hand, many solutions have drawbacks such as a deficiency in coercion resistance and receipt freeness, expenses associated with functioning on a public Blockchain, and susceptibility to certain types of assaults.

## Figures and Tables

**Figure 1 sensors-22-07585-f001:**
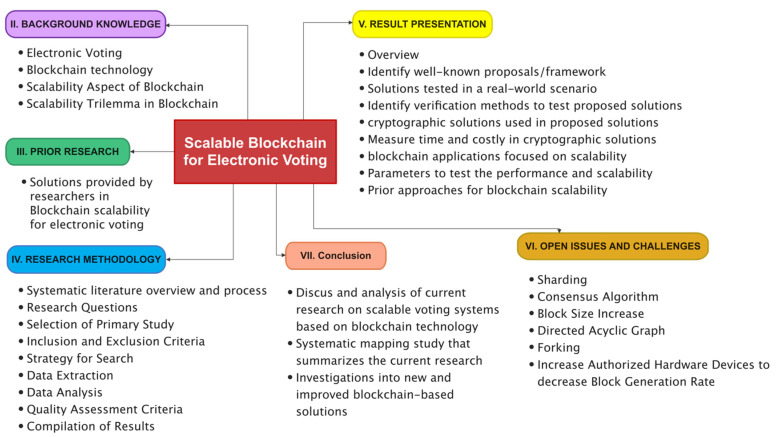
Road map of Blockchain-based electronic voting.

**Figure 2 sensors-22-07585-f002:**
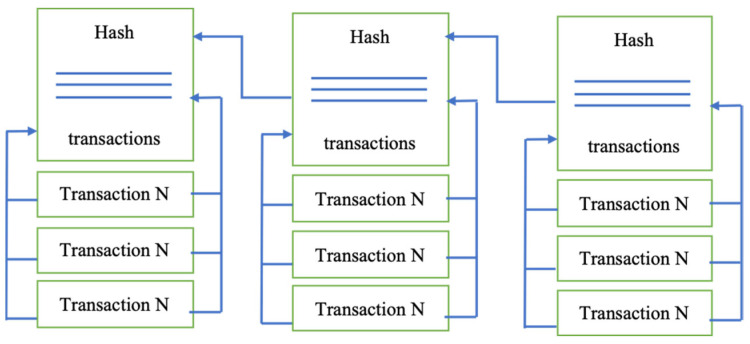
Model of a Blockchain data structure.

**Figure 3 sensors-22-07585-f003:**
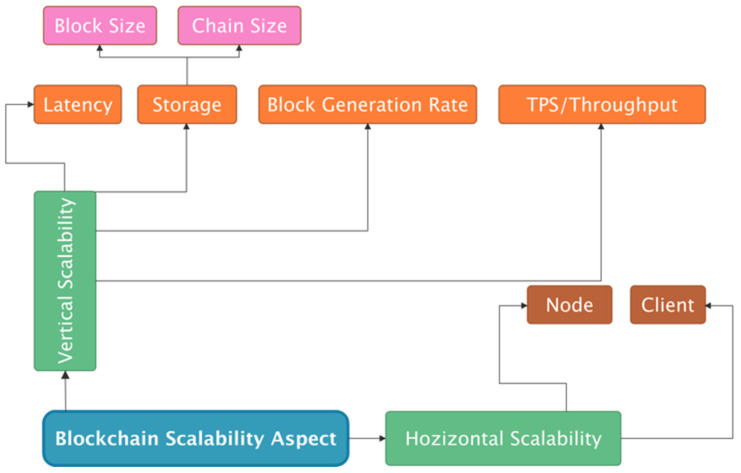
Blockchain scalability aspect.

**Figure 4 sensors-22-07585-f004:**
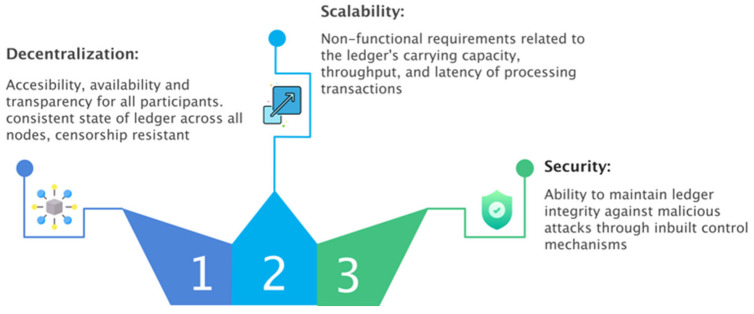
Blockchain Implications.

**Figure 5 sensors-22-07585-f005:**
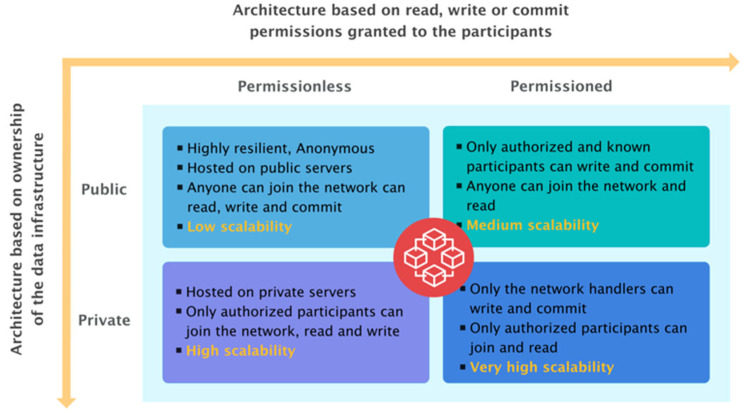
Blockchain scalability trilemma.

**Figure 6 sensors-22-07585-f006:**
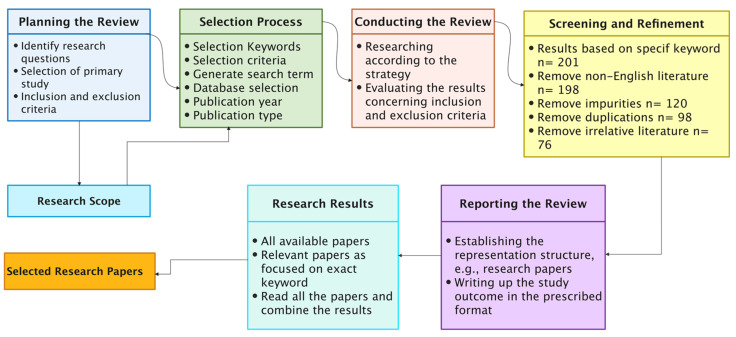
The process used to identify, gather, and refine research articles.

**Figure 12 sensors-22-07585-f012:**
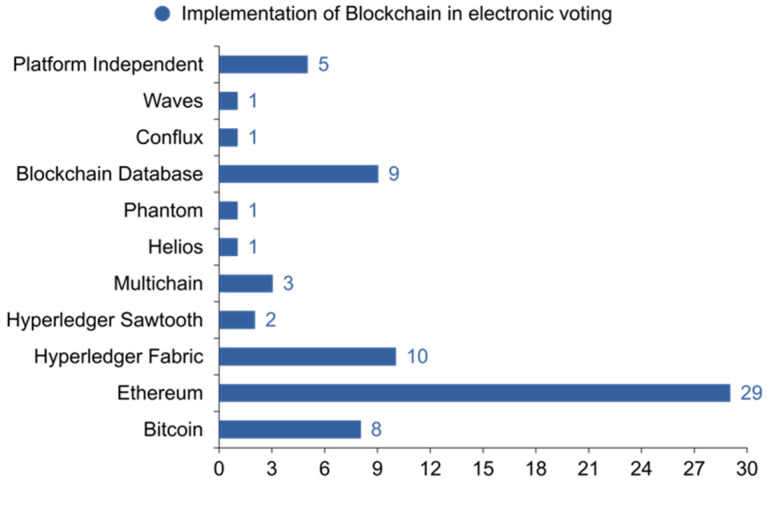
Implementation of Blockchain in electronic voting.

**Table 1 sensors-22-07585-t001:** Electronic voting system requirements.

Requirements	Description
Anonymity	A vote should not be associated with a voter.
Auditability and accuracy	Voting processes should be able to be checked, audited, and certifiable by autonomous agents.
Democracy/singularity	Every voter should be allowed to vote only once.
Vote privacy	There is no way to prove the voter by his/her casted vote.
Robustness and integrity	It should be impossible to change or eliminate votes after they have been cast.
Voter verifiability	Everyone should be able to independently confirm that all the votes have been tallied accurately.
Verifiable participation/authenticity	Only those voters who have the right to cast a vote are verified by the system.
Transparency and fairness	Voting systems should be transparent and rely on the accuracy, precision, and protection of voter security.
Availability and mobility	Voting systems should be permanently accessible during the election period. Voting systems should not restrict the voting location.
Accessibility and reassurance	Voting systems should be available for people with disabilities or special conditions without requiring specific equipment or abilities.
Recoverability and identification	Electoral systems can detect flaws, defects, and attacks and restore voting data to their previous state.

**Table 2 sensors-22-07585-t002:** Research questions table.

ID	Question	Description
RQ1	What are some of the most well-known proposals for scalable Blockchain-based electronic voting?	This topic highlights the most widely used Blockchain implementations utilized as the foundation for electronic voting systems. It enables the comparison and contrasting of several Blockchain and their attributes.
RQ2	Were those solutions tested in a real-world scenario?	It might be possible to improve upon existing solutions by looking at how they were implemented in the actual world.
RQ3	What are the verification methods used to test those solutions?	This question tries to find out how a solution was tested against the specifications of an electronic voting system.
RQ4	What are the different cryptographic solutions employed in previous research?	Many cryptographic primitives and procedures are used in today’s electronic voting systems. The primary objective of this investigation is to locate them so that information about them may be incorporated into other potential solutions.
RQ5	Were the cryptographic operations used in prior solutions too costly and time-consuming?	In the current scenario, cryptosystems are using too much power to solve the puzzle. Analysis of those solutions that claim to reduce the computational cost.
RQ6	What are the latest Blockchain applications focused on scalability?	Blockchain applications are no longer limited to Bitcoin. Understanding the real influence of Blockchain technology on scalability will need an examination of the most recent practical uses.
RQ7	What parameters test the performance and scalability of the electoral process on a large scale?	This question aims to test the current scalable Blockchain solutions on a large scale to check the performance of these solutions on a national-based electronic voting system.
RQ8	What are the prior approaches for Blockchain scalability to efficiently enhance the electoral process on a large scale?	This question aims to highlight the previous scalable solutions, approaches, and future directions on a national level.

**Table 3 sensors-22-07585-t003:** Query strings table.

Sr.	Query Strings
1.	(“blockchain” OR “block-chain” OR “distributed ledger”) AND “voting”
2.	(“blockchain” OR “block-chain” OR “distributed ledger”) AND (“evoting” OR “electronic voting” OR “Internet voting” OR “Ivoting”) AND “voting system”
3.	(“blockchain” OR “block-chain” OR “distributed ledger”) AND (“evoting” OR “electronic voting” OR “Internet voting” OR “Ivoting”) AND (“scalability” OR “scalable”) AND “large-scale”
4.	(“blockchain” OR “block-chain” OR “distributed ledger”) AND (“evoting” OR “electronic voting” OR “Internet voting” OR “Ivoting”) AND (“scalability” OR “scalable”) AND (“scalable” OR “scalability”) AND “national level”
5.	(Blockchain OR block-chain OR distributed ledger) AND (large-scale OR national level) AND (voting) AND (scalable OR scalability) AND (lightweight)

**Table 4 sensors-22-07585-t004:** Inclusion criteria for systematic literature review.

ID	Inclusion Criteria (IC)
IC1	Complete studies of at least five pages are required for this survey. It must be a peer-reviewed work that has been published in conference proceedings or journals.
IC2	Included studies must be published in the last five years (01.01.2017–31.03.2022).
IC3	Studies respond to a research question or propose a suitable solution about scalability in electronic voting.
IC4	Research examining the scalability of electronic voting systems based on Blockchain technology.
IC5	Empirical data relating to the application and Blockchain studies that address scalable Blockchain-based electronic voting must be included in the study.

**Table 5 sensors-22-07585-t005:** Exclusion criteria for systematic literature review.

ID	Exclusion Criteria (EC)
EC1	Those studies that are not written in English.
EC2	Studies that were published before (01.01.2017) and after (31.03.2022).
EC3	Studies that are irrelevant to the study since they do not address research questions.
EC4	Grey literature should not be included.
EC5	Duplicated studies.
EC6	Papers more than five years old.

**Table 6 sensors-22-07585-t006:** Data extraction.

Field	Description
ID	Individual study identifier.
Authors	List of the authors.
Year	Year of the journal.
Title	Title of the journal.
Source	IEEE Digital Library, Springer Link, Scopus, ACM Digital Library, and Science Direct
Type	Journal and conference papers only.
Research Question 1	What are some of the most well-known proposals for scalable Blockchain-based electronic voting?
Research Question 2	Were those solutions tested in a real-world scenario?
Research Question 3	What are the verification methods used to test those solutions?
Research Question 4	What are the different cryptographic solutions employed in previous research?
Research Question 5	Were the cryptographic operations used in prior solutions too costly and time-consuming?
Research Question 6	What are the latest Blockchain applications focused on scalability?
Research Question 7	What parameters test the performance and scalability of the electoral process on a large scale?
Research Question 8	What are the prior approaches for Blockchain scalability to efficiently enhance the electoral process on a large scale?

**Table 7 sensors-22-07585-t007:** Quality criteria.

ID	Quality Assessment Criteria (QAC)
QAC1	Does the research contribute to the field of Blockchain-based electronic voting that can be scaled up?
QAC2	Are the methods described in these papers being utilized in practice today?
QAC3	Are practitioners going to benefit from this article?
QAC4	Are the limitations of the proposed method presented and examined?
QAC5	Are the findings of the research being addressed in any way?
QAC6	Is there a suitable explanation provided for the background of the study?
QAC7	Is the context of the study sufficiently explained?
QAC8	Does the research include presentations of the related works?
QAC9	Are the study’s objectives and scope adequately stated?

**Table 8 sensors-22-07585-t008:** Papers about Blockchain for electronic voting selected for analysis.

ID	Article Title	Research Method	Research Result	Research Gap	Ref.
1	Secure large-scale E-voting system based on Blockchain contract using a hybrid consensus model combined with sharding	Simulation: Proposed PSC-Bchain hybrid consensus	Security, performance, and scalability of Blockchain-based e-voting systems	Coercion resistance and receipt freeness	[123]
2	COVID-19: Implementation e-voting Blockchain Concept	Quantitative: SUS trial analysis	Tackles election during COVID-19, brings effectiveness and efficiency	Authentication, refining the process of Blockchain for e-voting	[124]
3	On the Design and Implementation of a Blockchain Enabled E-Voting Application Within IoT-Oriented Smart Cities	Simulation: Designed a new system model for e-voting application	Privacy, trust, and security	Real-time data	[125]
4	Empirical analysis of transaction malleability within Blockchain-based e-Voting	Experimental: Process model diagram for transaction malleability	Network delay and block generation rate	Needs to develop new mechanisms and methods to mitigate malleability attack	[126]
5	Towards A Privacy-Preserving Voting System Through Blockchain Technologies	Experimental: Proposed algorithm for privacy-preserving voting	Political race control, control hacking of electronic democratic machine, cost-productive	Adaptability, coercion resistance, and scalability	[127]
6	Efficient, Coercion-free and Universally Verifiable Blockchain-based Voting	Experimental: Proposed algorithm with zkSNARK	Coercion resistance, receipt freeness, and universal verifiability	Trustworthy election authorities	[128]
7	Architecture-Centric Evaluation of Blockchain-Based Smart Contract E-Voting for National Elections	Architecture trade-off analysis method (ATAM)	Security attacks, internal vote manipulation, and endorsed transparency	Lack of security and consensus protocol	[129]
8	An enhanced security mechanism through Blockchain for E-polling/counting process using IoT devices	Simulation: Proposed e-voting layout with IoT devices	Enhanced security via Blockchain in e-voting applications using IoT devices	Insufficient evidence to handle malicious IoT devices	[130]
9	AttriChain Decentralized traceable anonymous identities in privacy-preserving permissioned Blockchain	Simulation: Proposed AttriChain protocol for security in permissioned Blockchain	Increases user privacy and autonomy in a permissioned network and enables auditing	Inability to deal with financial scenarios, public Blockchains are not supported	[131]
10	Blockchain voting Publicly verifiable online voting protocol without trusted tallying authorities	Simulation: Proposed a novel encryption scheme	Voters can store, verify, and tally all submitted votes	Double encryption and zero-knowledge/partial-knowledge proofs are weaknesses	[132]
11	A Blockchain-based self-tallying voting protocol in decentralized IoT	Simulation: Proposed self-tallying voting protocol with IoT	Self-tallying technology encrypts votes for a specified duration to guarantee voting confidentiality	Yes/no voting does not work in boardrooms or classrooms	[133]
12	Anonymous and Coercion-Resistant Distributed Electronic Voting	Simulation: Proposed scheme resistant to double voting	Enhanced the integrity, efficiency, and voter turnout of the election process	Ignores security and privacy	[134]
13	Blockchain Technology based Electoral Franchise	Simulation: Proposed architecture BCT-based electronic franchise	Immutable and distributable to minimize voting system assaults	Needs trusted third-parties, data integrity, and verification	[135]
14	A hyper-ledger fabric framework as a service for improved quality e-voting system	Proposed framework as a service	A highly maintainable, large-scale, cost-effective solution for personalized private Blockchain	Private Blockchain compromised data integrity and privacy	[136]
15	E-Voting System using Hyperledger Sawtooth	Experimental: Using Sawtooth with Solidity language	Large-scale implementation, reliability, as well as security	Nontransparent; system may be flawed based on consensus	[137]
16	Electronic voting based on virtual id of aadhar using Blockchain technology	Proposed framework: virtual ID based on UIDAI	Secured e-voting system by using biometric details	Generating and analysing fingerprints is hard	[138]
17	Blockchain-Based Self-Tallying Voting System with Software Updates in Decentralized IoT	Simulation: It uses IoT. The algorithm’s efficiency and average runtime are examined	A self-tallying voting system with software updates in decentralized IoT	Inability to deal with user privacy and security	[39]
18	Digital Voting A Blockchain-based E-Voting System using Biohash and Smart Contracts	Simulation: Smart contract and bio hash	Data integrity and anonymity, privacy, security	Fingerprint authentication is expensive and inaccurate	[139]
19	A Blockchain-based Traceable Self-tallying E-voting Protocol in AI Era	Simulation: AI-based self-tallying e-voting scheme	Satisfies anonymity, time-bounded privacy, linkability, and full traceability	Absence to real-world applications on large scale	[140]
20	Decentralized E-voting system based on Smart Contract by using Blockchain Technology	Simulation: Ethereum with smart contract on local blockchain	Reliable, safe, flexible, and able to support real-time services	Latency throughput issues arise. Ethereum’s speed limits large-scale deployment.	[141]
21	Efficient, coercion-free and universally verifiable Blockchain-based voting	Simulation: Ethereum with smart contract with a focus to reduce the gas fee	Scalable and practical for large-scale elections	A trustworthy administrator registers voters and an aggregator compile results	[128]
22	TrustVote On Elections We Trust with Distributed Ledgers and Smart Contracts	Simulation: Ethereum with Hyperledger	Compares permissioned and public Blockchain for better performance, transaction speed, and privacy	Coercion-resistant, needs trusted authorities	[46]
23	Efficient Privacy-Preserving Electronic Voting Scheme Based on Blockchain	Simulation: Ethereum with multicandidate electronic voting scheme	Good performance and the feasibility and correctness of the voting scheme	Permissioned Blockchain requires trusted authorities, scalability issues on large scale	[142]
24	Provotum A Blockchain-based and End-to-end Verifiable Remote Electronic Voting System	Simulation: End-to-end verifiable remote private Blockchain scheme	Practical design and architecture of public bulletin boards, ballot secrecy, and end-to-end verifiability	Does not support multiway elections, lack of secure communication channels	[143]
25	Investigating performance constraints for Blockchain based secure e-voting system	Simulation: Permissioned and permissionless blockchain to measure performance	Efficiency, performance, and scalability	It does not explicitly state security aspects such as uniqueness and ballot reception	[1]
26	A Smart Contract System for Decentralized Borda Count Voting	Simulation: A decentralized, multicandidate, public-verifiable voting system.	Self-tallying decentralized and ranked-choice voting system	Large-scale voting is also impossible. No protocol considers voter privacy	[144]
27	Secure Online Voting System Using Biometric and Blockchain	Experimental: Using Biometric with sidechain	Tamper-proof storage, user authenticity, and data security	It cannot engage stakeholders on Blockchain quality or feasibility	[14]
28	An improved FOO voting scheme using Blockchain	Experimental: many-functional Blockchain-based FOO protocol	Fairness and correctness	Coercion-resistant	[145]
29	Fault-Tolerant Architecture Design for Blockchain-Based Electronics Voting System	Simulation: fault-tolerant e-voting systems	Data security, performance, measuring computing resources and agility	Weaknesses in anonymization, Internet security, and managing unforeseen events	[146]
30	Large-Scale Electronic Voting Based on Conflux Consensus Mechanism	Simulation: comparative Blockchain-based voting systems	Decentralized, self-managed, noninteractive, and free of charge	Inability to deal coercion-resistant privacy and security	[147]
31	Securing e-voting based on Blockchain in P2P network	Simulation: using Python with ECC public key cryptography	Avoids forgery of votes, authentication, nonrepudiation, and changing of vote before a preset deadline	The system’s PKI database may invalidate its vote block verification process	[24]
32	Towards A Privacy-Preserving Voting System Through Blockchain Technologies	Proposed: Cost-productive framework	Fairness, independence, and unbiasedness	The proposed system was not tested and implemented	[148]
33	End-to-End Voting with Non-Permissioned and Permissioned Ledgers	Simulation: E2E verifiable system using Bitcoin and MultiChain	Uncoercibility and receipt freeness and data confidentiality and neutrality	Not formally proven, needs to measure the qualitative properties	[149]
34	An Anti-Quantum E-Voting Protocol in Blockchain with Audit Function	Simulation: Niederreiter’s code-based cryptosystem	True fairness, audit voters, transparency, and efficiency in small-scale election	Largely appropriate for small-scale elections	[150]
35	Chaintegrity: Blockchain-enabled large-scale e-voting system with robustness and universal verifiability	Simulation: Impractical properties of Blockchain-based e-voting	Large-scale e-voting system	Uniqueness and ballot reception are not explicitly stated	[151]
36	voteChain: Community Based Scalable Internet Voting Framework	Proposal presented: Mixed approach to achieve require results	Scalable Internet voting	Multiple vote casting, possibility of attack	[152]
37	A Secure Decentralized Trustless E-Voting System Based on Smart Contract	Simulation: Privacy-preserving e-voting protocol	Decentralized, trustless electronic voting system based on the smart contract	Cannot scale, needs a lot of processing and communication power	[153]
38	Towards Blockchain-Based E-Voting Systems	Simulation: Efficiency- and scalability-based system	Cost-effective, lightweight, and efficient	The study did not specify the system’s needs	[34]
39	Votereum: An Ethereum-Based E-Voting System	Simulation: Robust, private, and verifiable system	Privacy, uniqueness, universal verifiability, and robustness	This mechanism cannot assure free receipt or withstand coercion	[154]
40	TrustedEVoting (TeV) a Secure, Anonymous and Verifiable Blockchain-Based e-Voting Framework	Conceptual design: Secure and verifiable e-voting system (TeV)	Nontampering, voter anonymity, and vote verifiability	TeV was not implemented or evaluated in a nationwide election	[29]
41	LaT-Voting: Traceable Anonymous E-Voting on Blockchain	Simulation: Traceability prefixes anonymous protocol	Authentication, privacy, and subtle traceability	No central authority to monitor	[155]
42	A Simple Voting Protocol on Quantum Blockchain	Simulation: Quantum Blockchain electronic	Anonymous, binding, nonreusable, verifiable, eligible, fair, and self-tallying	It lacks auditability consistency	[156]
43	Transparent E-Voting dApp Based on Waves Blockchain and RIDE Language	Simulation: Privacy and tally voting system	Substantially enhances performance	Waves votes dApps. Large-block, voter-controlled blockchains	[157]
44	Design of Blockchain-based electronic election system using Hyperledger: Case of Indonesia	Proposed design: Blockchain necessity, issue solution, and secure election needs	Security and performance	Security and performance were not tested on the design	[158]
45	DABSTERS: A Privacy Preserving e-Voting Protocol for Permissioned Blockchain	Formal Verification: Election privacy and transparency protocol	Voter privacy, integrity, and verifiability	ProVerif validated. Performance and implementation assessments are lacking	[159]
46	Privacy-protected Electronic Voting System Based on Blockchin and Trusted Execution Environment	Simulation: Cost-effective and efficient electronic voting	Economical, efficient, and privacy for board-scale elections	Needs more research on large-scale elections and the system	[26]
47	Implementation of an E-Voting Scheme Using Hyperledger Fabric Permissioned Blockchain	Simulation: Consortium blockchain with smart contracts	Increased transparency and trust	The system was developed for general e-voting, not stakeholder demands	[160]
48	Designing Process-Centric Blockchain-Based Architectures A Case Study in e-voting as a Service	Architecture Design: Blockchain-based e-voting with cloud	Process-centric, trusted, configurable and multipurpose electronic voting service	Security and privacy were not discussed and it was not implemented	[161]
49	Blockchain centered homomorphic encryption: A secure solution for E-balloting	Simulation: Blockchain voting through additive homomorphism	Prevention of multiple voting, privacy and end-to-end verifiability	Scalability and multicasting voting issues	[162]
50	A privacy-preserving voting protocol on Blockchain	Simulation: Homomorphic encryption counts smart-contract votes	End-to-end privacy, detectability, and correctability against cheating	The system is unfair and does not prevent malicious activity	[163]
51	SHARVOT: Secret SHARe-Based VOTing on the Blockchain	Proposed Solution: Used circle shuffle technique	Privacy, anonymity, ballot irrevocability, and transparency	Needs centralization. Authorities sabotage voting	[164]
52	Improving end-to-end verifiable voting systems with Blockchain technologies	Experimental: Helios has DDoS and data manipulation difficulties	End-to-end verifiability, data tampering, immutability, and public accessibility	This voting technique involves a “centralized database service”	[165]
53	SecEVS: Secure Electronic Voting System Using Blockchain Technology	Experimental: Secure electronic voting system using Blockchain technology	Transparency, decentralization, irreversibility, and nonrepudiation	This work is costly and time-consuming	[166]
54	A proposal of Blockchain-based electronic voting system	Experimental: double-envelope encryption and Blockchain	High availability and universal verifiability	E-voting privacy was not discussed	[7]
55	PHANTOM protocol as the new crypto-democracy	Mathematical Modelling: PHANTOM protocol with encription method	Vote rigging, hacking, election manipulation, and polling booth capturing	This protocol was not implemented or evaluated	[167]
56	Blockchain-based e-voting system	Experimental: Scheme uses a smart contract to tally the result	Security and decreased cost of hosting a nationwide election	Compromised security, privacy, and anonymity	[168]
57	E-Voting with Blockchain: An E-Voting Protocol with Decentralisation and Voter Privacy	Experimental: Proposed a decentralized voting scheme	Open, fair, and independently verifiable	It does not provide auditability, consistency, or privacy	[169]
58	Blockchain-Enabled E-Voting	Experimental: Smartphone-based electronic platform	Reduced voter fraud and increased voter access	It is more of an example of i-voting rather than e-voting	[38]
59	Crypto-voting, a Blockchain based e-Voting System	Experimental: Voting uses sidechains, holds voters and vote data	Traceability and auditability	Compromising on fairness	[170]
60	Towards secure e-voting using ethereum Blockchain	Experimental: Voting through Ethereum with Android application	Safer, cheaper, more secure, more transparent, and easier to use	The drawbacks are that it lacks robustness and receiving freedom	[171]
61	Towards the intelligent agents for Blockchain e-voting system	Conceptual model: Auditable Blockchain voting system multiagent concept	Transparency and auditability	Data and privacy insecurity via email voting	[53]
62	A Survey of Blockchain Based on E-voting Systems	Survey: It highlights current security and privacy vulnerabilities	Security and privacy requirements of the existing Blockchain e-voting systems	It lacks scalability, throughput, and latency	[53]
63	E-Voting on the Blockchain	Analysis: Blockchain voting is discussed	Reduced voter fraud and increased voter access	It is not entirely decentralized; utilizes Blockchain to store votes	[172]
64	A study on decentralized e-voting system using Blockchain technology	Proposed model: E-voting system with SHA voter data encryption	Vote rigging and hacking of the EVM	One chain per candidate adds storage and processing complexity	[173]
65	A secure end-to-end verifiable e-voting system using zero knowledge based Blockchain	Proposed scheme: DRE-ip with zero-knowledge-based public Blockchain	Authenticated, end-to-end verifiable, and with secret ballot election	Large computational costs make these technologies costly to use and deploy in DeFi.	[174]
66	Voting process with Blockchain technology: Auditable Blockchain voting system	Proposed scheme: supervised Internet voting system	Audit and verification capability	Lack of audits and system verification compromises transparency and tamper-proofness	[175]
67	Secure Digital Voting System Based on Blockchain Technology	Scheme: E-voting using jeopardizing cryptographic hashes	Increased efficiency and reduced errors	It allowed multiple votes to one user	[176]
68	Decentralized Voting Platform Based on Ethereum Blockchain	Experimental: Voting on multiple votes per mobile (MSISDN)	Ensuring data integrity, transparency, enforcement, and privacy	Absence of clarity and time consumption	[177]
69	Blockchain based e-voting recording system design	Simulation: Recording of voting result using Blockchain algorithm	Security, transparency, and reduced cheating sources of database manipulation	Lack of personal voter authentication	[178]
70	Decentralized E-Voting Systems Based on the Blockchain Technology	Scheme: Blockchain-based, secret sharing and homomorphic encryption	Anonymity, privacy, and verifiability	Missed quality targets	[179]
71	An e-voting protocol based on Blockchain	Scheme: Blockchain and blind signature	Satisfied coercion resistance and access control	The limits are coercion resistance, robustness, and fairness	[180]
72	The future of e-voting	Academic proposal: Zcash protocol with zero knowledge	Security and anonymity	A server challenge-response handshake to stores email, fake emails are simple	[181]
73	A conceptual secure Blockchain-based electronic voting system	Academic proposal: voting scheme using hash value	Secure, reliable, anonymous voting might improve local or national elections	Lack of authentication and privacy for users	[182]
74	A smart contract for boardroom voting with maximum voter privacy	Academic proposal: Comprehensive secure protocol using the Ethereum	An Internet voting system that is both decentralized and self-tallying while also protecting the privacy of voters	Their arithmetic-heavy technique requires a lot of computing power	[183]
75	E-Voting System Based on Blockchain Technology: A Survey	Analysis: Blockchain voting security and transparency	Less trust, privacy, and security	Absence of dialogue around scalability and anonymity	[184]
76	Blockchain for Electronic Voting System Review and Open Research Challenges	Analysis: Blockchain voting challenges and future research	Privacy protection, scalability, and transaction speed	Usage authentication not stated	[11]

**Table 10 sensors-22-07585-t010:** The methods of verification employed in the chosen articles.

Verification Method	Research Article
Examination of evaluation criteria	[175,184]
Scalability testing (nodes, transactions per second, response time)	[182,185]
Performance tests (blocks per hour, blocks per day, transaction latency, transaction throughput)	[123,176]
Security test/analysis (vulnerability towards specific types of attacks)	[125,181]
Cost evaluation	[127,171]

**Table 11 sensors-22-07585-t011:** Explicitly revealed cryptographic solutions in researched articles.

Cryptography	Cryptography Solution	Research Article
Symmetric Key	Advanced Encryption Standard (AES)	[127,134]
Asymmetric/Public Key	Homomorphic	[128,179]
	RSA	[137,160]
	Elliptic Curve Cryptography (ECC)	[24,162]
	Elliptic Curve Digital Signature Algorithm (ECDSA)	[129,181]
	ElGamal	[132,162]
	DSA	[144]
Hash Functions	SHA’s	[182,185]

**Table 12 sensors-22-07585-t012:** Cryptographic operations.

Cryptography Solution	Articles	Cost	Time (Key Generation Time)
Advanced Encryption Standard (AES)	[127,134]	N/A	1 KB file size encryption time: 128-bit key generation ~14,594 microsecond, 192-bit key generation ~14,432 microsecond, and 256-bit key generation ~16,222 microsecond. Relatively fast over all ~ 0.0023 s.
Homomorphic	[128,179]	N/A	Bits slower ~ 41 milliseconds.
RSA	[137,160]	512-bit RSA keys—2 CPU hours (the cost of USD 0.06) 1024-bit RSA keys—97 CPU days (the cost of USD 40–80) 2048-bit RSA keys—140.8 CPU years, (the cost of USD 20,000–USD 40,000)	1024-bit key generation ~0.16 s, 2240-bit key generation ~7.47 s, 3072-bit key generation ~9.80 s. Overall key generation time ~15 min, RSA bits slower.
Elliptic Curve Cryptography (ECC)	[24,162]	160 bits ECC keys— (the cost of USD 0.02) 224 bits ECC keys— (the cost of USD 3333.33–6666.67)	1024-bit key generation ~0.08 s, 2240-bit key generation ~0.18 s, 3072-bit key generation ~0.27 s.
Elliptic Curve Digital Signature Algorithm (ECDSA)	[129,181]	N/A	163-bit key generation ~0.08 s, 233-bit key generation ~0.18 s, 283-bit key generation ~0.27 s, 409-bit key generation ~0.64 s, 571-bit key generation ~1.44 s,
ElGamal	[132,162]	N/A	25 KB file size encryption time: ~1858.6 millisecond, 50 KB file size encryption time: ~2346.2 millisecond.
Digital Signature Algorithm (DSA)	[144]	N/A	512-bit key generation ~19 millisecond, 1024-bit key generation ~82 millisecond.
SHA’s (SHA-2 to SHA 256)	[182,185]	N/A	1.4 MB file size 0.04 s, 2.2 MB ~ 0.07 s, 3.3 MB ~0.13 s, 5.9 MB ~ 0.21 s and 6.9 MB ~ 0.24 s.

**Table 13 sensors-22-07585-t013:** Scalability scenario classification.

Scalability Scenario Classification	Research Articles
Small Scale or Organization Voting	[183,185]
National Voting	[123,167]
Generic Voting	[135,161]

**Table 14 sensors-22-07585-t014:** Identified performance parameters.

Performance Parameter	Description	Articles
Block frequency	Interval time between two succeeding blocks	[140,178]
Block size	A block’s capacity to hold the number of transactions	[24,185]
Workload type	Smart contract	[141,165]
CPU Usage/Node configuration	CPU, network speed, RAM	[14,34,179]
Network size	The number of nodes	[181,186]
Network structure/Network usage	Network structure of the Blockchain	[125,182]
Workload quantity	Total transactions to be carried out	[1,165]
Amount of miners/sealers/Memory usage	Actively participating nodes	[14,164]
Blockchain client and API	E.g., Geth or Parity, web3.js or web3.py	[136,166]

**Table 15 sensors-22-07585-t015:** Identified scalability parameters.

Scalability Metric	Description	Articles
Throughput	Number of accomplished transactions per second (TPS)	[123,178]
Latency	The time difference in seconds between when a transaction is submitted and completed	[126,163]
Scalability	When a parameter is altered, there are changes in both throughput and latency (e.g., the network size or the hardware configuration of a node)	[147,177]

## Data Availability

Not applicable.

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
