# Peer review of "A Systematic Literature Review and Meta-Analysis on Scalable Blockchain-Based Electronic Voting Systems"

_sensors, 2022, doi:10.3390/s22197585_

Round 1

Reviewer 1 Report

Thanks for submitting the paper.

Please double check the copyright of all figures, especially Figure 9.

Author Response

Thanks for reviewing our manuscript. As per the review, We have doubled check all figures in this paper. All the figures are copyright free, created, and drawn as per our concept.

Reviewer 2 Report

Dear Authors,

The purpose of present paper is to review and analyze the current research on scalable voting systems, primarily based on Blockchain technology by using SLR based on Preferred Reporting Items for Systematic Reviews and Meta-Analyses (PRISMA).

Many thanks for submitting the paper to this journal and for the time you spent writing that. There are some comments that will help you improve your research;

1. In the abstract section; the time period for the selection of studies and the number of databases used to extract papers should be added to this section. Also, the research results should be clearly mentioned.

2. The number of keywords is more than usual. For example, Internet of things is the same as IOTs. Please check again and reduce if possible.

3. As much as the second part of paper is written with many explanations, the third section (related work) is short and concise. It is better to prepare an organized table including a number of related studies along with the references, research methods used, and a summary of their results in section 3 and then point out the research gap.

4. In Line 160; it is better to mention the abbreviation (VVAPTs) right after the word. Also, in Line 879; DSA can be used instead of Digital Signature Algorithm, since the word abbreviation is mentioned in line 813. Please check the entire text for abbreviations.

5. Each of the tables or figures that are taken from the reference, the reference should be added to their caption.

6. Although Figure 2 is included in the paper (Page 8), it is not referred in the text. Please check it out.

7. Please re-check the numbering of subsections, for example, 2.3 has been repeated twice.

8. In Lines 441; “DAO” is the abbreviation of which word?

9. The explanations provided in lines 468-470 is repeated again in lines 478-479, which can become one.

10. In Fig.6; check the screening and refinement box.

11. It should be clear in the diagram how many papers have been removed at each stage of the screening process.

12. The concern about the adopted methodology is related to the qualitative approach. In particular, the review approach described in paper is based on a subjective classification of the papers, that cannot be supported by an objective quantitative analysis but consider several papers "irrelevant" just according to the reviewers' opinion. Therefore, the concern is that different reviewers would classify and select different papers that could lead to different conclusions. The robustness of the adopted methodology should be better explained.

13. Are the opinions of the experts used in the screening process of the papers or authors? If experts' opinions are used, the demographic characteristics of the experts should be mentioned.

14. Please check the text carefully for grammatical errors to improve the readability.

15. PRISMA checklist should be added to the end of the paper.

Author Response

Thanks for reviewing our manuscript and providing great suggestions. As per the review, We have made all changes in our manuscript outlined below:

  1. In the abstract section: we have made all the changes like period, the number of databases and research results.
  2. Number of Keywords: We have reduced the number of keywords as suggested
  3. Related Work: This SLR is based on Blockchain-based voting, so we have chosen 76 articles per our selected query after screening and refinement closely related to the topic. Then we analyzed the research gap in section 5 (Result Presentation) with research methodology and the results in Table 8 and improved the table. If we increase more, the length of the paper will be exceeded.
  4. Text Abbreviations: Checked all abbreviations.
  5. Tables or Figures Reference: All the tables and figures are copyright free, created, and drawn as per our concept.
  6. Cross-reference: Checked and Cross-referenced figure 2
  7. Numbering of subsections: We have checked and corrected the numbering of subsections.
  8. "DAO" Abbreviation: We have added the abbreviation.
  9. Repeated text: Removed repeated text
  10.  Fig 6: Checked and made changes as per suggestion
  11. Screening Process: Update figure 6 and mention how many papers have been removed at each stage.
  12.  Adopted Methodology: This SLR design aims to develop a way to show the final results consistently when the process is subject to various "voting methods." To achieve systematic efforts about the scalability of blockchain networks in electronic voting, we have systematically combined the results of multiple studies in an appropriate way to get better results. This systematic review included studies with numerical data, but we have not driven this SLR to be more technical than we have shaped with more subjectivity to the topic. We have clearly formulated questions that use systematic and explicit methods to identify, select and critically appraise relevant to this research research, and to extract and analyze data from the studies included in this review. However, please let us know if you have more suggestions to improve the methodology.
  13. Screening process opinion: The screening process is based on the author's choice and the demand of the research area. 
  14. Grammatical errors: Checked all grammatical errors and improved readability
  15. PRISMA checklist: We will added the PRISMA checklist as a supporting document.

Reviewer 3 Report

This manuscript provides Literature Review on Scalable Blockchain based  Electronic Voting Systems.

Consider following:

- Explain on which criteria, research questions are selected in Table 2 ? These points could also be covered in general.

- Length of paper needs to be reduced. Authors should briefly describe about background, instead of providing details for example table 4,5,6 and other similar details.  

- Paper should be more focused on technical details and discussion

- Table 8 shows benefits only, also mention weaknesses and discuss it in discussion section.

Author Response

Thanks for reviewing our manuscript and providing great suggestions. As per the review, We have made all changes in our manuscript outlined below:

  1. Research Questions Criteria: As per our research, we interpret what we found in the literature and check if it is feasible, riveting, novel, ethical, and relevant to this research. During our examination of the literature, we ensured that the questions we produced were sufficient to demonstrate their relevance and that the techniques of analysis used were suitable. Yes, it can be covered except for Q1, Q7 and Q8.
  2. Paper Length and Table explanation: Table 4,5,6 is already explained in their sub-sections like inclusion, exclusion criteria and data extraction. As per our thinking no need to explain more to those tables because our primary focus is on the results derived from the study.
  3. Focused Technical Details: We tried our best to explain every question with a table and technical details, but we cannot go deep because it will go out of the scope of this research and questions.   
  4. Table 8:  As per the suggestions, we have added two more columns in table 8, research gap and research method.

Round 2

Reviewer 2 Report

All of my comments were done by authors.

Reviewer 3 Report

I am ok with updated version.

Thanks